# Increasing ambient temperature progressively disassemble *Arabidopsis* phytochrome B from individual photobodies with distinct thermostabilities

Joseph Hahm[1,2], Keunhwa Kim[1,2], Yongjian Qiu [1] & Meng Chen [1]✉

Warm temperature is postulated to induce plant thermomorphogenesis through a signaling mechanism similar to shade, as both destabilize the active form of the photoreceptor and thermosensor phytochrome B (phyB). At the cellular level, shade antagonizes phyB signaling by triggering phyB disassembly from photobodies. Here we report temperature-dependent photobody localization of fluorescent protein-tagged phyB (phyB-FP) in the epidermal cells of *Arabidopsis* hypocotyl and cotyledon. Our results demonstrate that warm temperature elicits different photobody dynamics than those by shade. Increases in temperature from 12 °C to 27 °C incrementally reduce photobody number by stimulating phyB-FP disassembly from selective thermo-unstable photobodies. The thermostability of photobodies relies on phyB's photosensory module. Surprisingly, elevated temperatures inflict opposite effects on phyB's functions in the hypocotyl and cotyledon despite inducing similar photobody dynamics, indicative of tissue/organ-specific temperature signaling circuitry either downstream of photobody dynamics or independent of phyB. Our results thus provide direct cell biology evidence supporting an early temperature signaling mechanism via dynamic assembly/disassembly of individual photobodies possessing distinct thermostabilities.

[1] Department of Botany and Plant Sciences, Institute for Integrative Genome Biology, University of California, Riverside, CA 92521, USA. [2] These authors contribute equally: Joseph Hahm, Keunhwa Kim. ✉email: meng.chen@ucr.edu

Temperature is a critical environmental cue influencing all aspects of plant development and growth. Besides the stress-tolerant responses induced by extreme freezing or scorching conditions[1–3], changes in the range of ambient growth temperatures (12–27 °C for *Arabidopsis*[4]) can also dramatically alter the rate of shoot and root growth and thereby plant architecture[5,6], adjust cell differentiation patterns and developmental transitions such as reproductive organ formation[7–9], and modulate plant immunity[10–12]. These phenomena are collectively called thermomorphogenesis[13,14]. Increases in global temperature have already had major impacts on plant phenology, distribution, diversity, and are expected to significantly decrease crop productivity[15,16]. Therefore, a molecular understanding of how plants sense and respond to temperature has become critical to the development of strategies for sustaining crop production in a changing climate[17].

Plants monitor changes in ambient temperature by the red (R) and far-red (FR) photoreceptor, phytochrome B (phyB)[18,19]. PhyB is a bilin-containing pigment or a biliprotein that senses light using a covalently linked phytochromobilin as the chromophore[20–22]. Light induces the isomerization of the C15-C16 carbon–carbon double bond of phytochromobilin between a C15-Z (*trans*) and a C15-E (*cis*) configurations, and whereby, photoconverts phyB between two relatively stable conformations: an inactive, R light-absorbing Pr (C15-Z) and an active, FR light-absorbing Pfr (C15-E)[20–22]. The photoconversion of phyB, through changes in distinct ratios between active Pfr to inactive Pr, allows plants to detect changes in light quality, quantity, and periodicity, thereby sensing the diurnal and seasonal time as well as the physical environment. For example, the existence of neighboring plants or vegetative shade enriches FR light and shifts the phyB equilibrium towards its inactive Pr form to induce the shade avoidance response[23–25]. The Pfr/Pr equilibrium of phyB can also be influenced by temperature[18]. The active Pfr of phyB is thermally unstable and can spontaneously revert back to the inactive Pr in a light-independent process called dark or thermal reversion[20,26,27]. The thermal reversion of phyB is rapid and can be greatly enhanced by increases in temperature between 10 and 30 °C[18]. This temperature-dependent, intrinsic property makes phyB a thermosensor of ambient temperature[18,19]. Therefore, temperature signaling by phyB is thought to be mediated by a similar mechanism as that of shade[14,18,19].

Ambient temperature signaling is best studied in *Arabidopsis* (*Arabidopsis thaliana*) by examining how temperature modulates phyB-mediated light responses[13,14]. PhyB exerts prominent roles in photomorphogenesis during *Arabidopsis* seedling development, wherein photoactivation of phyB triggers differential organ growth of the embryonic stem (hypocotyl) and the embryonic leaf (cotyledon)[28]—that is, while phyB promotes cotyledon expansion, it restricts hypocotyl elongation[29,30]. Hypocotyl elongation has been widely used as a readout for measuring phyB activity. Exposing *Arabidopsis* seedlings to warm temperatures (e.g., a transition from 21 to 27 °C) results in enhanced hypocotyl elongation akin to shade treatments[13,14,31]. In both scenarios, shifting phyB's equilibrium towards the inactive Pr releases its inhibitory effects on the stability and activity of Phytochrome-Interacting Factors, the PIFs, which are a small group of nodal basic/helix–loop–helix transcriptional regulators promoting plant growth through the activation of genes involved in the biosynthesis and signaling of the plant growth hormone auxin and other hormones[13,14,32]. In contrast to hypocotyl elongation, cotyledon (and also leaf) expansion is promoted by phyB[29]. The current model of phyB's inhibitory role on the growth-promoting PIFs does not offer an intuitive explanation for phyB-mediated cotyledon expansion. Nonetheless, seedlings exposed to shade conditions display smaller cotyledons, which is consistent with the idea that shade inhibits phyB activity[33]. The temperature effect on phyB-mediated cotyledon expansion has not been explicitly described.

One of the earliest light responses at the cellular level—by either activation or inactivation of phyB—is the alteration of phyB's subcellular distribution[34]. Photoactivation of phyB promotes its assembly to subnuclear compartments named photobodies (PBs), while inactivation of phyB triggers its disassembly from PBs[28,35–37]. The light-dependent phyB assembly to PBs has been extensively studied using *Arabidopsis* transgenic lines expressing functional phyB tagged with a fluorescent protein (phyB-FP). PhyB-FPs are synthesized in the dark as the Pr form in the cytoplasm. During the dark-to-light transition, when seedlings emerge from the soil and encounter light, photoactivated phyB-FPs translocate from the cytoplasm to the nucleus and initially localize to many small foci, and several hours later, to a few large foci referred hereafter as PBs[36–38]. Conversely, during the light-to-dark transition, the inactivation of phyB-FP by dark/thermal reversion stimulates its redistribution from PBs to small foci and the nucleoplasm[39]. The steady-state pattern of phyB-FP in continuous light also alters with changes in light quantity and quality (i.e., R/FR ratio) and correlates tightly with phyB-mediated regulation of hypocotyl growth in *Arabidopsis*[40]. Previous studies of the steady-state PB dynamics by molecular genetics and computational modeling support the notion that PB localization requires the Pfr[36,40–43]. Strong light conditions, which stabilize the Pfr, promote the assembly of phyB-FP PBs of 0.7–2 μm in diameter[40,44]. By contrast, shade conditions and dim light, which enhance the conversion to the Pr, lead to the relocalization of phyB-FP from PBs to tens of small foci of 0.1–0.7 μm in diameter and the nucleoplasm[40,45]. Molecular genetic studies on mutants with altered PB patterns indicate that PBs are required for phyB signaling[44,46–53].

Changes in ambient temperature can also alter the morphology of PBs. It has been shown that transitions from 20 °C to different temperatures between 10 and 30 °C altered the size of PBs quadratically—that is, PBs were the largest at 20 °C and became smaller in either cooler or warmer temperatures[18]. However, the current data do not explain why PBs responded to temperature quadratically. It is also unclear, since warm temperatures act similarly as shade[18,19], whether warm temperatures induce similar PB dynamics as shade. Moreover, because phyB triggers differential responses in hypocotyl and cotyledon growth, it remains unknown whether temperature induces different PB dynamics in early phyB signaling in the hypocotyl and cotyledon. To address these questions, we characterized the steady-state PB patterns of phyB-FP in epidermal cells from the hypocotyl and cotyledon of two *Arabidopsis* ecotype backgrounds, Col-0 and L*er*, in four ambient growth temperatures, 12, 16, 21 and 27 °C. We show that increases in temperature from 12 to 27 °C progressively reduce the number of PBs in all cell types by stimulating phyB-FP disassembly from selective thermo-unstable PBs, while thermostable PBs persist even at 27 °C. Although the PB formation is mediated by phyB's C-terminal module, the thermostability of PBs depends on phyB's N-terminal photosensory module. Surprisingly, cotyledon and hypocotyl cells display similar PB dynamics despite opposing temperature effects in phyB signaling, indicating tissue/organ-specific temperature signaling mechanisms downstream of PB dynamics. These results thus demonstrate that temperature induces largely different PB dynamics than those by shade in early phyB signaling, thereby providing direct cell biology evidence supporting a unique temperature signaling mechanism via dynamic assembly and disassembly of individual PBs with distinct thermostabilities.

## Results

**Tissue/organ-specific temperature effects on phyB signaling.**
PBs have been characterized extensively using transgenic lines
expressing phyB-FP[36,39,40,44–46,54,55], and therefore to examine
the dynamics of phyB-FP PBs induced by temperature changes,
we went back to the *phyB-FP* lines, including *PBG* (*phyB-GFP*),
which expresses functional phyB-GFP to complement the null
*phyB-5* mutant in the Landsberg *erecta* (L*er*) background[36], and
*PBC* (*phyB-CFP*), which expresses phyB-CFP to complement the
null *phyB-9* mutant in Col-0[46,55]. Because both *PBG* and *PBC*
overexpress phyB-FP[36,55], we first examined whether they retain
the same temperature effects as their respective ecotype back-
ground. In response to temperature increases from 12 °C to 27 °C,
L*er* and Col-0 seedlings grown in continuous 10 μmol m$^{-2}$ s$^{-1}$ R
light exhibited gradual increases in hypocotyl length in a 2.4- and
4.5-fold dynamic range, respectively (Fig. 1a, b)[56]. *PBG* and *PBC*
lines showed reduced hypocotyl responses with a 1.4-fold
dynamic range for both lines. However, they retained similar
temperature effects on phyB signaling as L*er* and Col-0, respec-
tively—that is, warmer temperatures attenuated the function of
phyB or enhanced hypocotyl elongation in all four lines (Fig. 1a, b).
The reduced temperature responses in hypocotyl elongation in
*PBG* and *PBC* are likely due to overexpression of phyB-FPs.
Increasing the flux of phyB signaling in L*er* and Col-0 did not
lead to a similar effect as overexpression of phyB-FPs. For
example, L*er* and Col-0 seedlings grown in a higher light inten-
sity, 50 μmol m$^{-2}$ s$^{-1}$ R light (R50), showed even greater
dynamic ranges of the temperature response (Fig. 1a, b), sug-
gesting that the dynamic range of the temperature response in
hypocotyl elongation relies on the level of phyB. Together, these
results indicate that *PBG* and *PBC* maintain a reduced but similar
warm temperature-dependent antagonistic effect on phyB-
dependent inhibition of hypocotyl elongation as L*er* and Col-0,
respectively.

We next examined the effect of temperature on cotyledon
expansion. Seedlings of a variety of *Arabidopsis* ecotypes,
including L*er* and Col-0, accelerate cotyledon expansion when
moved from low intensity (i.e., below 10 μmol m$^{-2}$ s$^{-1}$) to high
intensity (i.e., above 50 μmol m$^{-2}$ s$^{-1}$) of R light, and this
cotyledon response depends on phyB[29,57]. Therefore, in contrast
to restricting hypocotyl growth, phyB promotes light-dependent
cotyledon expansion[29]. To our surprise, warmer temperatures
further enhanced phyB-dependent cotyledon expansion in L*er*
and Col-0 in permissive 50 μmol m$^{-2}$ s$^{-1}$ R light (Fig. 1c, d). We
did not observe a linear trend for the temperature effects on
cotyledon expansion in 10 μmol m$^{-2}$ s$^{-1}$ R light, despite a slight
reduction of cotyledon size from 16 to 27 °C (Fig. 1c, d).
Interestingly, *PBG* and *PBC* grown in 10 μmol m$^{-2}$ s$^{-1}$ R light
also showed increases in cotyledon area with temperature in a
similar manner as their respective ecotype-background lines
grown in 50 μmol m$^{-2}$ s$^{-1}$ R light (Fig. 1c, d). These results
suggest that the high levels of phyB-FP in *PBG* and *PBC* lines
could compensate for the low-light conditions and enable the
temperature-dependent cotyledon expansion response. Therefore,
we conclude that *PBG* and *PBC* retain the same temperature
effects in promoting phyB-dependent cotyledon expansion as L*er*
and Col-0, respectively.

**Temperature-dependent PB dynamics in the hypocotyl.** To
investigate how ambient temperature influences PB morphology,
we first characterized the steady-state patterns of phyB-GFP PBs
in epidermal-cell nuclei of the top one-third of the hypocotyl in
4-day-old *PBG* seedlings grown at 12, 16, 21, and 27 °C. We chose
to grow seedlings in 10 μmol m$^{-2}$ s$^{-1}$ R light because it has been
previously shown that in this light condition phyB-FP localizes

only to PBs (as opposed to some small foci as well)[40,55]. We
quantified the number and volume of PBs using three-
dimensional imaging analysis (see Methods). To our surprise,
we did not observe a shade-like transition from PBs to small foci
with temperature increases. Instead, phyB-GFP localizes to PBs in
all four temperatures (Fig. 2a, top panels). Intriguingly, the
average number of PBs per nucleus decreased progressively with
temperature increases from four-to-five at 12 °C to two-to-three
at 27 °C (Fig. 2b). We found that PBs can be categorized into two
types based on their relative positions to the nucleolus: one type
at the nucleolar periphery, referred hereafter as nucleolar-
associated PB (Nuo-PB), and the other type away from the
nucleolus, referred hereafter as non-nucleolar-associated PB
(nonNuo-PB). The numbers of both Nuo-PBs and nonNuo-PBs
decreased with temperature increases (Fig. 2b). The average
number of Nuo-PBs per nucleus dropped from two at 12 °C to
one at 27 °C (Fig. 2a, b). Similarly, the number of nonNuo-PBs
declined from two-to-three per nucleus at 12 °C to one per
nucleus at 27 °C (Fig. 2a, b). Consistently, the percentage of nuclei
containing two or more either Nuo-PBs or nonNuo-PBs reduced
gradually with temperature increases (Fig. 2c, d). For instance, at
12 °C, 87 ± 3% of the nuclei had two or more nonNuo-PBs,
whereas at 27 °C, the majority of the nuclei (67 ± 5%) had either
only one or zero nonNuo-PB (Fig. 2a, d). The volume of PB
became larger with temperature increases (Fig. 2e). Interestingly,
the amount of phyB-GFP in *PBG* increased with temperature and
was more than tripled at 27 °C compared to that at 12 °C (Fig. 2f).
Therefore, the increases in PB size with temperature could be due
to the redistribution of phyB-GFP to fewer PBs and/or the
enhanced phyB accumulation. Because the expression of *phyB-
GFP* was under the control of the 35S constitutive promoter[36], the
accumulation of phyB-GFP is likely due to a mechanism at the
post-transcriptional level. Together, these results show that
increases in temperature progressively reduce the number of PBs
in the nuclei of hypocotyl epidermal cells. Not all PBs reacted to
temperature equally, while some thermo-unstable PBs dis-
appeared in warmer temperatures, certain PBs appear to be
thermostable even at 27 °C.

To test whether the temperature-dependent PB dynamics vary
in different *Arabidopsis* ecotypes, we characterized the PBs in the
*PBC* line. Similar to *PBG*, the numbers of total, Nuo, and nonNuo
PBs in the nuclei of hypocotyl epidermal cells in *PBC* decreased
with temperature increases (Fig. 2a, b). The percentage of nuclei
containing two or more either Nuo-PBs or nonNuo-PBs declined
with temperature increases (Fig. 2a–d). The PBs also became
larger with temperature increases in *PBC*, so as the level of phyB-
CFP (Fig. 2e, f). The main difference between *PBC* and *PBG* was
that *PBC* had significantly more PBs per nucleus than *PBG*. This
difference was particularly obvious in lower temperatures. For
example, at 12 °C, while *PBG* contained four to five total PBs per
nucleus, *PBC* had almost twice as many (Fig. 2a, b). Interestingly,
while the numbers of Nuo-PBs of *PBG* and *PBC* differed only
slightly, the number of nonNuo-PBs per nucleus in *PBC* was
more than doubled compared with that in *PBG*—that is, two to
three nonNuo-PBs in *PBG* vs. six to eight in *PBC* (Fig. 2a, b).
Therefore, the differences in PB numbers between *PBC* and *PBG*
were mainly contributed by the numbers of nonNuo-PBs. The
differences in PB number between *PBG* and *PBC* were unlikely
caused by a difference in the amount of phyB-FP because the
levels of phyB-GFP and phyB-CFP are comparable between these
two lines across the temperature range examined (Fig. 2f).
Intriguingly, the dynamic changes in PB number by temperature
were also mainly contributed by the variations in nonNuo-PBs
(Fig. 2b), suggesting that nonNuo-PBs are more dynamically
regulated by both environmental and genetic factors. Together,
these results show that temperature changes trigger similar PB

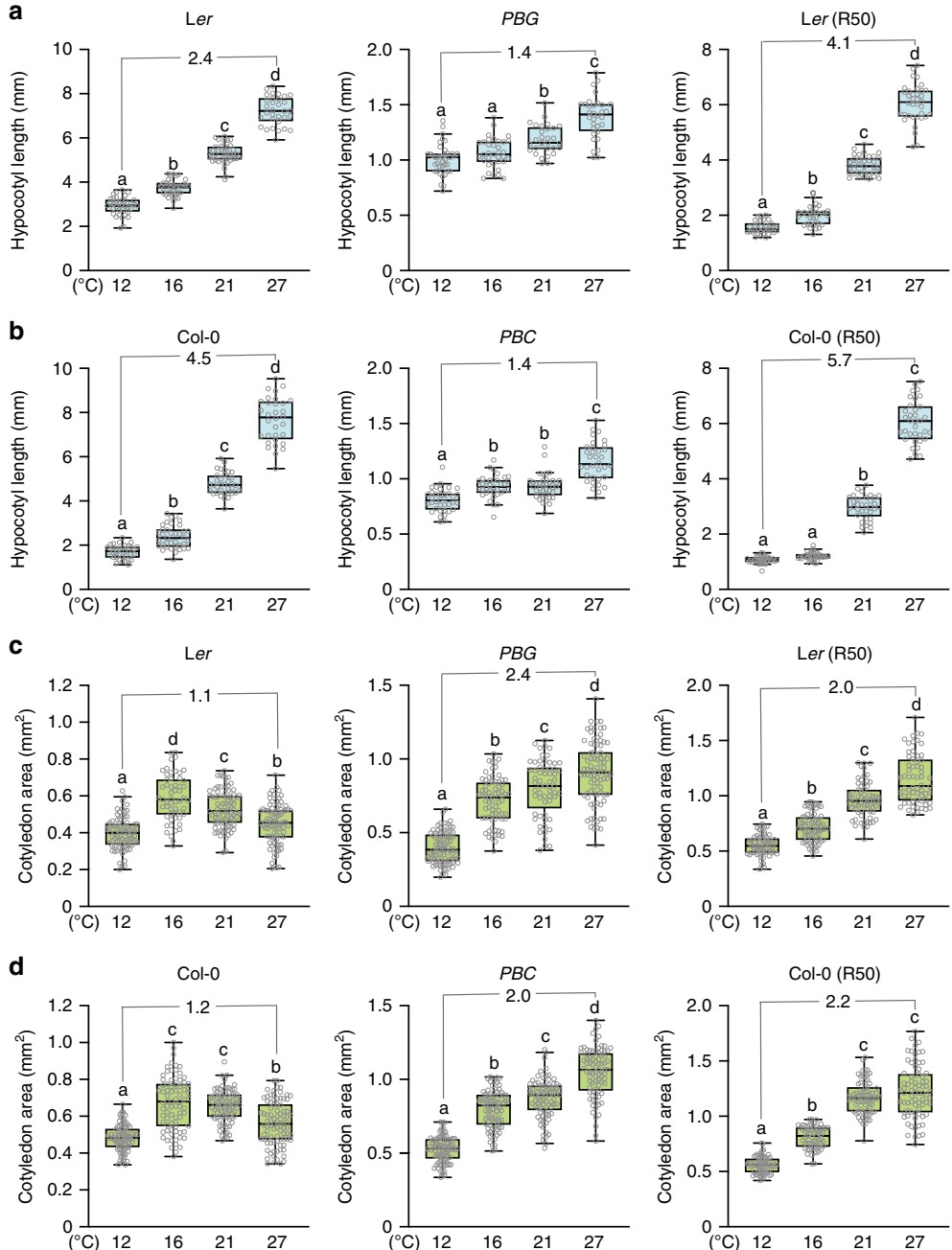

**Fig. 1 Temperature imposes opposite effects on phyB functions in hypocotyl and cotyledon. a, b** Warmer temperatures attenuate the function of phyB signaling in inhibiting hypocotyl growth. Hypocotyl length measurements of 4-day-old seedlings of L*er* and *PBG* (**a**) and Col-0 and *PBC* (**b**) grown under 10 or 50 (R50) µmol m$^{-2}$ s$^{-1}$ R light at 12, 16, 21, and 27 °C. **c, d** Warmer temperatures enhance the function of phyB signaling in promoting cotyledon expansion. Cotyledon size measurements of 4-day-old seedlings of L*er* and *PBG* (**a**) and Col-0 and *PBC* (**b**) grown under 10 or 50 (R50) µmol m$^{-2}$ s$^{-1}$ R light at 12, 16, 21, and 27 °C. For **a–d**, the fold changes of the median values of either hypocotyl length or cotyledon area between 27 and 12 °C are shown. For the box and whisker plots, the boxes represent from the 25th to the 75th percentile and the bars equal the median values. Samples with different letters exhibit statistically significant differences (ANOVA, Tukey's HSD, $P \leq 0.05$, $n \geq 33$). The source data underlying the hypocotyl (**a, b**) and cotyledon (**c, d**) measurements are provided in the Source Data file.

dynamics in the hypocotyl epidermal cells in L*er* and Col, despite variations in the precise number of PBs per nucleus between the two ecotypes. These results suggest that individual PBs vary in thermostability, and temperature changes trigger dynamic assembly/disassembly of selective thermo-unstable PBs, while thermostable PBs can persist even in elevated temperatures.

**Temperature-dependent PB dynamics in the cotyledon.** PB dynamics have been characterized mainly in epidermal cells of

the hypocotyl. It remains unclear whether PB dynamics are influenced by tissue/organ type. Given temperature elicits opposing effects on the function of phyB signaling in hypocotyl elongation and cotyledon expansion, we asked whether temperature induces similar or distinct PB dynamics in the cotyledon. To that end, we characterized the steady-state PB patterns under the same series of ambient temperatures in the epidermal cells on the adaxial side of the cotyledon. Interestingly, similar to hypocotyl epidermal cells, temperature increases from 12 to 27 °C led

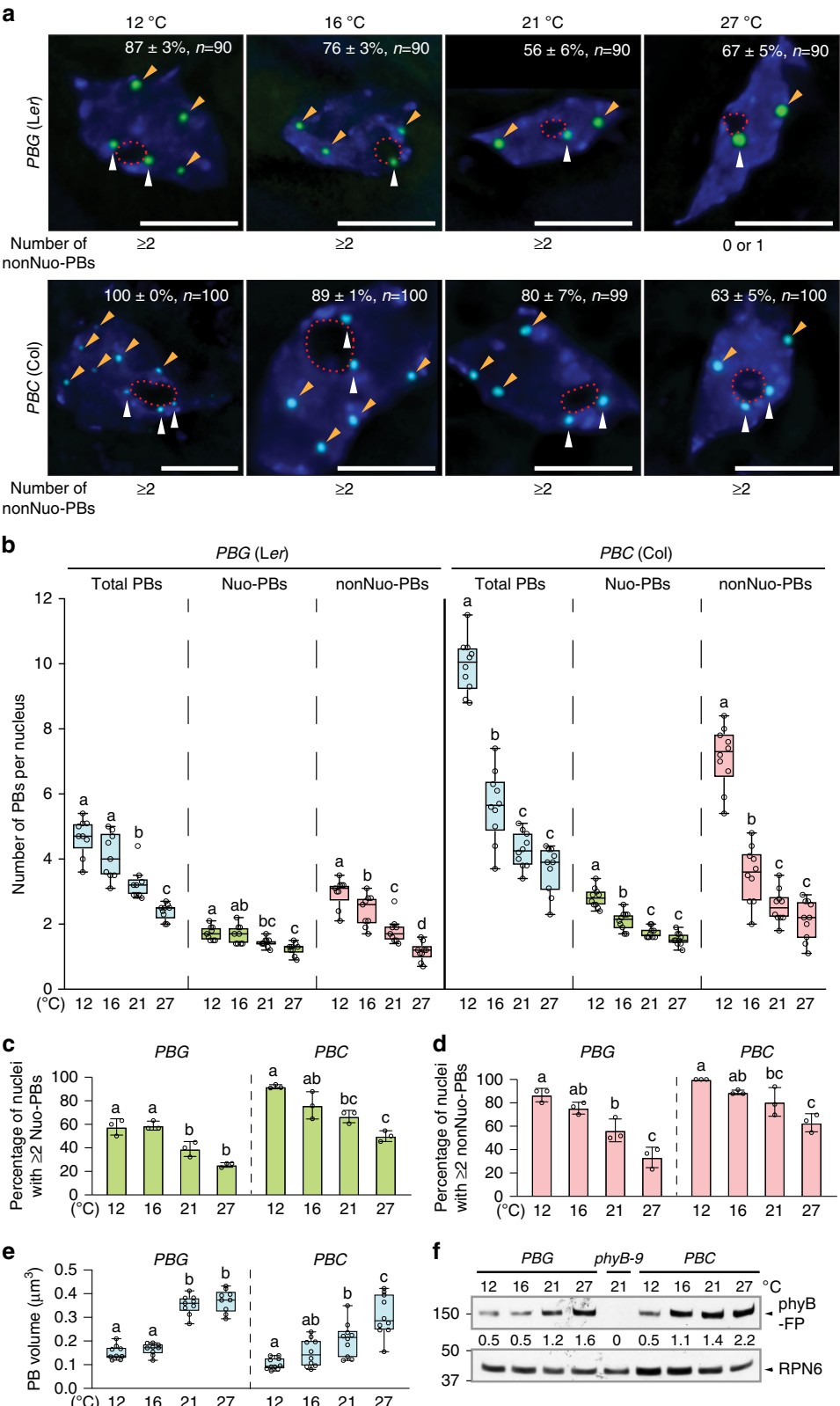

to progressive decreases in the total number of PBs in the cotyledon epidermal nuclei in both *PBG* and *PBC* (Fig. 3a–c). Notably, the cotyledon nuclei contained fewer PBs compared with hypocotyl nuclei. For example, the cotyledon nuclei in *PBG* grown at 12 °C had mostly two-to-three PBs vs. four-to-five PBs in the hypocotyl nuclei (Figs. 2b and 3b). Similarly, the cotyledon nuclei of *PBC* grown at 12 °C contained five-to-six PBs compared

to nine-to-ten PBs in the hypocotyl nuclei (Figs. 2b and 3c). In *PBC*, the numbers of both Nuo-PBs and nonNuo-PBs in cotyledon cells also declined with temperature increases (Fig. 3c); the percentage of nuclei with two or more of either Nuo-PBs or nonNuo-PBs decreased dramatically from 12 to 27 °C (Fig. 3a, d). Comparing the PB numbers between *PBG* and *PBC*, we found that the cotyledon epidermal nuclei in *PBC* had more PBs than

**Fig. 2 Temperature-dependent photobody dynamics in hypocotyl epidermal cells. a** Maximum-projection, deconvolved fluorescence microscopy images showing representative steady-state patterns of phyB-GFP (green) or phyB-CFP (cyan) PBs in hypocotyl epidermal-cell nuclei of 4-day-old *PBG* or *PBC* seedlings, respectively, grown under 10 µmol m$^{-2}$ s$^{-1}$ R light at 12, 16, 21 °C, and 27 °C. Nuclei were labeled by DAPI (blue), and the boundaries of the nucleoli are traced by red dashed lines. Nuo-PBs and nonNuo-PBs are indicated by white and orange arrowheads, respectively. The percentage of nuclei including s.e. with the indicated PB pattern is shown in each image; *n* indicates the total number of nuclei analyzed. Scale bars represent 5 µm. **b** Quantification of the numbers of total, Nuo, and nonNuo PBs per nucleus in hypocotyl epidermal-cell nuclei of the *PBG* and *PBC* seedlings described in **a**. **c** Quantification of the percentage of nuclei in hypocotyl epidermal-cell nuclei of the *PBG* and *PBC* seedlings described in **a** with two or more Nuo-PBs. **d** Quantification of the percentage of nuclei in hypocotyl epidermal-cell nuclei of the *PBG* and *PBC* seedlings described in **a** with two or more nonNuo-PBs. For **c**, **d**, error bars represent s.e. calculated from groups of seedlings. Different letters denote statistically significant differences (ANOVA, Tukey's HSD, $P \leq 0.05$, $n = 3$). **e** Quantification of the volume of PBs in hypocotyl epidermal-cell nuclei of the *PBG* and *PBC* seedlings described in **a**. For **b**, **e**, the boxes represent from the 25th to the 75th percentile and the bars equal the median values. Samples with different letters exhibit statistically significant differences (ANOVA, Tukey's HSD, $P \leq 0.05$, $n \geq 9$). **f** Immunoblot results showing the levels of phyB-GFP or phyB-CFP in 4-day-old *PBG* or *PBC* seedlings grown under 10 µmol m$^{-2}$ s$^{-1}$ R light in the indicated temperatures. phyB-FPs were detected by anti-phyB antibodies. RPN6 was used as a loading control, and *phyB-9* was used as a negative control. The relative levels of phyB-FPs, normalized against the corresponding levels of RPN6, are shown below the phyB-FP immunoblot. The source data underlying the PB measurements in **b**–**e**, and the immunoblots in **f** are provided in the Source Data file.

those in *PBG* (Fig. 3a–c), this trend is the same as the hypocotyl (Fig. 2a, b). Interestingly, we found that the majority of the nuclei in *PBG* cotyledon epidermal nuclei contained only two PBs—one Nuo-PB and one nonNuo-PB—at 12 °C ($68 \pm 3\%$), 16 °C ($64 \pm 3\%$), and 21 °C ($63 \pm 3\%$) (Fig. 3a, e). In striking contrast, at 27 °C, the nonNuo-PB disappeared in $67 \pm 3\%$ of the cotyledon epidermal nuclei in *PBG*, leaving only one Nuo-PB (Fig. 3a, b, e). More than 85% of cotyledon epidermal nuclei in *PBG* maintained one Nuo-PB across the temperature range between 12 and 27 °C (Fig. 3e), indicating distinct thermostabilities between the thermosensitive nonNuo-PB and the thermo-insensitive Nuo-PB. The reduction of PB number with temperature increases suggests that the thermostability of PBs is determined by the stability of the Pfr form of phyB, which is attenuated by warmer temperatures[18]. Consistent with this hypothesis, increasing light intensity from 10 to 50 µmol m$^{-2}$ s$^{-1}$ R light, which stabilizes the Pfr, increased the numbers of PBs in the cotyledon cells of both *PBG* and *PBC* lines at 27 °C (Fig. 3a–d). Similar to hypocotyl PBs, the size of the cotyledon PBs also increased with temperature increases (Fig. 3f). Together, these results further support the notion that individual PBs exhibit distinct thermostabilities. The number of PBs can be influenced by tissue/organ-specific factors. The fact that cotyledon and hypocotyl epidermal cells display similar PB dynamics suggests that the opposing effects of temperature on phyB signaling might be caused by tissue/organ-specific signaling circuitry downstream of PB dynamics or independent of phyB.

**Temperature-induced rapid disassembly of thermosensitive PBs.** The simplicity of the PB pattern in the *PBG* cotyledon epidermal nuclei allows us to distinguish the two PBs based on their positions to the nucleolus, and therefore provides an opportunity to determine the kinetics of the disappearance of the thermosensitive nonNuo-PB during the 21 to 27 °C transition. A major challenge in determining PB dynamics is that the localization pattern of phyB-FPs can be rapidly and dramatically altered by the excitation light during live-cell fluorescence imaging[58]. To circumvent this problem, we fixed the seedling samples and characterized the steady-state phyB-GFP PB patterns at selected time points within 12 h after the 21 to 27 °C transition (Fig. 4a). The results show that the percentage of nuclei containing one nonNuo-PB dropped from about 60% at 21 °C at time 0 to 36% after 6 h at 27 °C and maintained at this percentage for the rest of the time course (Fig. 4a, b). In contrast, the percentage of nuclei containing one Nuo-PB did not change (Fig. 4c). Neither the level of phyB-GFP nor the size of PBs varied more than 2-fold (Fig. 4d, e). Together, these results demonstrate that individual PBs could respond to a specific temperature range and

temperature increases induce rapid disassembly of phyB-GFP from distinct thermosensitive PBs.

**phyB's C-terminal module localizes to thermo-insensitive PBs.** The structural basis of the light-dependent PB localization of phyB has been extensively investigated. PhyB is a homodimer; PB localization requires the dimeric Pfr form of phyB[40,41,43,46,59]. Each phyB monomer contains an N-terminal photosensory module and a C-terminal output module[20,21]. PB localization of phyB is mediated by the dimeric form of phyB's C-terminal output module, which presumably contains localization signals or sequences for targeting phyB to the nucleus and PBs[46,54,55]. PhyB's C-terminal module alone localizes constitutively to PBs[54,55]. The current model posits that the subcellular targeting activities of the C-terminal output module is masked in the Pr form by the N-terminal module, at least in part, through light-dependent interactions between the two modules, the interaction between N- and C-terminal modules weakens in the Pfr and thereby the PB localization activity of the C-terminal module is unmasked or exposed[55]. If this model is correct, the C-terminal module alone would be expected to localize to PBs that are temperature insensitive and should show a similar pattern to that of active phyB. To test this hypothesis, we used the *BCY* line, which expresses the phyB C-terminal module fused with YFP (BCY) in the *phyB-9* background[55], and tested whether the localization of BCY to PBs in cotyledon-cell nuclei in the dark can be altered by temperature changes between 12 and 27 °C. Supporting our hypothesis, the BCY-containing PBs did not respond to temperature changes. In all four temperatures tested from 12 to 27 °C, most of the nuclei contained more than eight PBs, including at least two Nuo-PBs and six nonNuo-PBs (Fig. 5a, b). The percentage of nuclei with such a PB pattern stayed the same across the temperature range (Fig. 5d). Interestingly, comparing the PB pattern of *BCY* with that of *PBC* (both in the *phyB-9* background), the pattern of BCY PBs resembles that of PBC PBs at 12 °C, where the Pfr is more stabilized (Fig. 3a, c). The number of BCY PBs, including Nuo-PBs and nonNuo-PBs, also stayed the same between dark- and light-grown *BCY* lines (Fig. 5a, b, d). These results indicate that the localization of BCY to PBs is not responsive to light and temperature, and also is not influenced by possible heterodimerization with other phys in the light[60,61]. Consistent with the thermo-insensitivity of BCY PBs, the cotyledon size of *BCY* did not respond to temperature changes in the dark (Fig. 5e). The sizes of dark-grown *BCY* cotyledons were at least 5-fold smaller than those of *PBC* grown in the light, suggesting that phyB's C-terminal module alone is not sufficient in promoting cotyledon expansion (Figs. 1d and 5e). *BCY* seedlings grown in the light did show significantly larger cotyledons

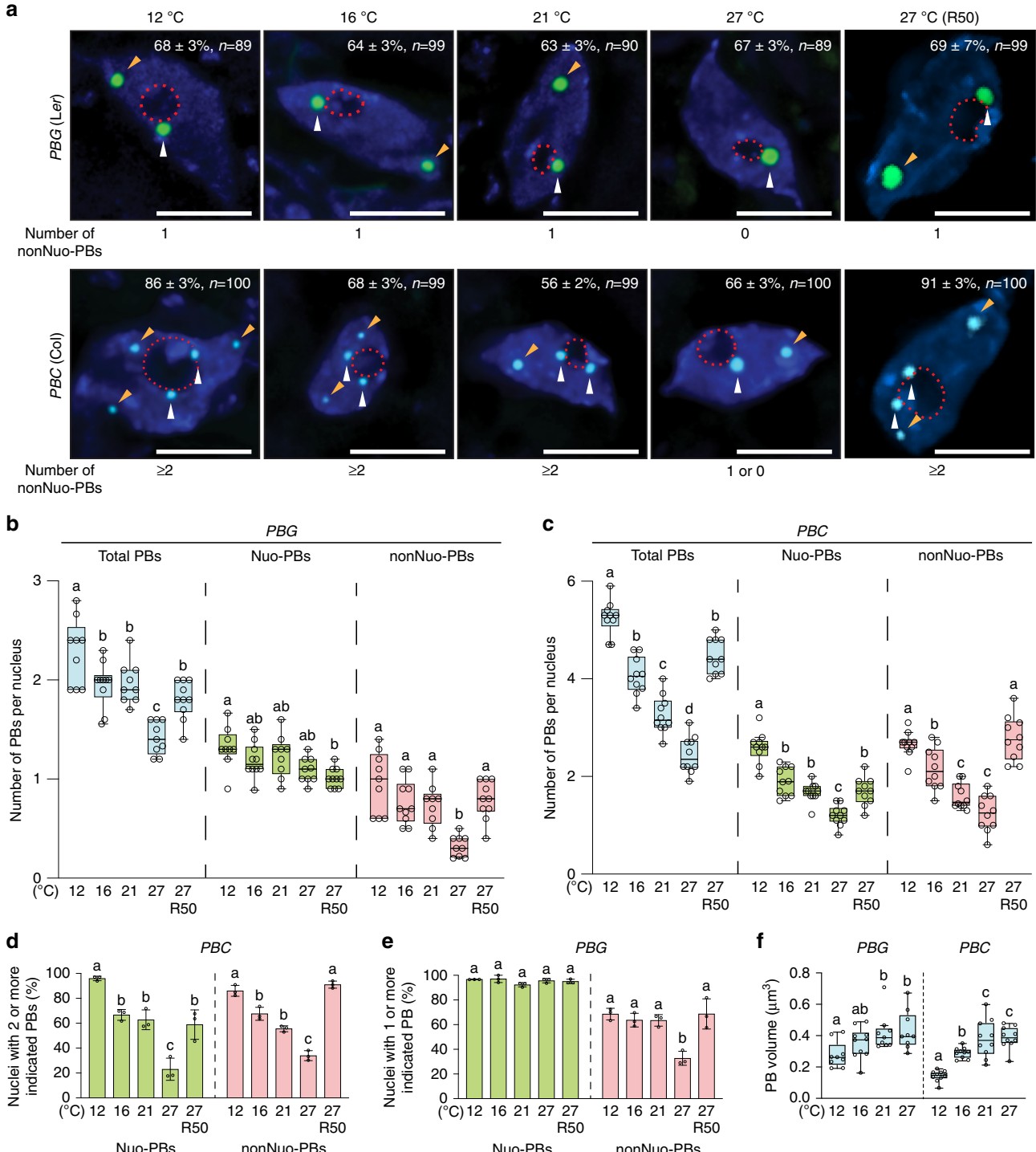

**Fig. 3 Temperature-dependent photobody dynamics in cotyledon epidermal cells. a** Maximum-projection, deconvolved fluorescence microscopy images showing representative steady-state patterns of phyB-GFP (green) or phyB-CFP (cyan) PBs in cotyledon epidermal-cell nuclei of 4-day-old *PBG* or *PBC* seedlings, respectively, grown under 10 μmol m$^{-2}$ s$^{-1}$ R light at 12, 16, 21, and 27 °C, as well as under 50 μmol m$^{-2}$ s$^{-1}$ R light at 27 °C (R50). Nuclei were labeled by DAPI (blue), and the boundaries of nucleoli are traced by red dashed lines. Nuo-PBs and nonNuo-PBs are indicated by white and orange arrowheads, respectively. The percentage (with s.e.) of nuclei with the indicated PB pattern is shown in each image; *n* indicates the total number of nuclei analyzed. Scale bars represent 5 μm. **b**, **c** Quantification of the numbers of total, Nuo, and nonNuo PBs per nucleus in cotyledon epidermal-cell nuclei of the *PBG* (**b**) and *PBC* (**c**) seedlings described in **a**. **d** Quantification of the percentages of nuclei in cotyledon epidermal-cell nuclei with two or more either Nuo- or nonNuo-PBs in the *PBC* seedlings described in **a**. **e** Quantification of the percentage of nuclei in cotyledon epidermal-cell nuclei containing one or more either Nuo- or nonNuo-PB in the *PBG* seedlings described in **a**. For **d**, **e**, error bars represent s.e. calculated from groups of seedlings. Different letters denote statistically significant differences (ANOVA, Tukey's HSD, *P* ≤ 0.05, *n* = 3). **f** Quantification of the volume of PBs in cotyledon epidermal-cell nuclei of the *PBG* and *PBC* seedlings described in **a**. For **b**, **c**, **f**, the boxes represent from the 25th to the 75th percentile, and the bars equal the median values. Samples with different letters exhibit statistically significant differences in PB number (ANOVA, Tukey's HSD, *P* ≤ 0.05, *n* ≥ 9). The source data underlying the PB measurements in **b**–**f** are provided in the Source Data file.

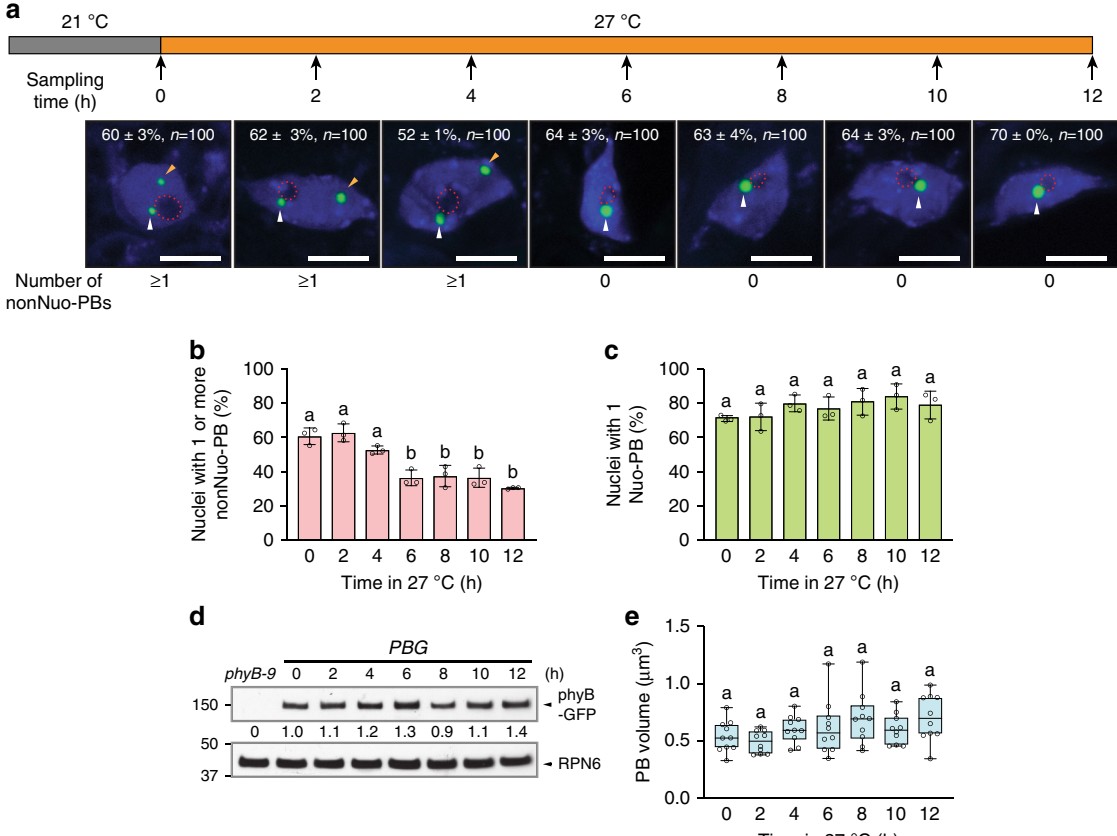

**Fig. 4 Kinetics of warm temperature-induced disappearance of the nonNuo-PB in *PBG*. a** Schematic illustration of the 21 to 27 °C transition experiment with maximum-projection, fluorescence microscopy images showing representative steady-state patterns of phyB-GFP PBs in cotyledon epidermal-cell nuclei of 4-day-old *PBG* seedlings at the indicated time points. Nuclei were labeled by DAPI (blue), and the boundaries of the nucleoli are traced by red dashed lines. Nuo-PBs and nonNuo-PBs are indicated by white and orange arrowheads, respectively. The percentage of nuclei with s.e. showing the indicated PB pattern is included in each image; *n* indicates the total number of nuclei analyzed. Scale bars represent 5 μm. **b** Quantification of the percentage of cotyledon epidermal-cell nuclei with one or more nonNuo-PB in the *PBG* seedlings described in **a**. **c** Quantification of the percentage cotyledon epidermal-cell nuclei with one Nuo-PB in the *PBG* seedlings described in **a**. For **b**, **c**, error bars represent s.e. calculated from groups of seedlings. Different letters denote statistically significant differences (ANOVA, Tukey's HSD, $P \le 0.05$, $n = 3$). **d** Immunoblot results showing the levels of phyB-GFP in 4-day-old *PBG* seedlings grown under 10 μmol m$^{-2}$ s$^{-1}$ R light in the indicated time points during the 21 to 27 °C transition. phyB-GFP was detected by anti-phyB antibodies, and RPN6 was used as a loading control. *phyB-9* was used as a negative control. The relative levels of phyB-GFP normalized against the corresponding levels of RPN6 are shown below the phyB-GFP immunoblot. **e** Quantification of the volume of PBs in cotyledon epidermal-cell nuclei of the *PBG* seedlings during the 21 to 27 °C transition described in **a**. The boxes represent from the 25th to the 75th percentile, and the bars equal the median values. Samples with the same letter exhibit no statistically significant difference in PB volume (ANOVA, Tukey's HSD, $P \le 0.05$, $n = 10$). The source data underlying the PB measurements in **b**, **c**, **e**, and the immunoblots in **d** are provided in the Source Data file.

compared with dark-grown *BCY* seedlings (Fig. 5e), which is likely due to the actions of other phys. The level of BCY also increased with temperature increases (Fig. 5f). However, in this case, the volume of individual BCY PBs did not change significantly (Fig. 5g), possibly due to the distribution of the additional BCY to the large numbers of PBs. Together, these results indicate that temperature does not affect the PB localization activity of the C-terminal module of phyB per se, and therefore the thermostability of PBs likely depends on phyB's N-terminal photosensory module[18].

**YHB preferentially localizes to one thermo-insensitive PB.** To further examine the structural basis of PB's thermostability, we turned to the constitutively active *phyB* mutant YHB, which carries a Y276H mutation in phyB's chromophore attachment domain and locks phyB in an active form[53]. It is important to note that although YHB is biologically active, it represents a unique active conformation because YHB is poorly photoactive

and stuck mainly in an R light-absorbing conformation—that is, a biologically active Pr form[53,62–64]. Although YHB has been shown to localize constitutively to PBs even in the dark[50,53,65], the pattern of YHB PBs has not been characterized in comparison with the wild-type phyB PBs. If the thermostability of PBs is determined by the N-terminal module, a constitutively active phyB is expected to show a similar PB pattern as BCY or phyB-FP in low temperatures. To test this hypothesis, we used a *YHB* line, which expresses YHB-YFP in the *phyB-9* background[66]. The pattern of YHB-YFP PBs in cotyledon epidermal nuclei in the dark stayed the same between 12 and 27 °C (Fig. 5a, c). However, this pattern was strikingly different from that of BCY. The majority of the nuclei contained only a YHB-YFP Nuo-PB without any nonNuo-PB, and the rest nuclei contained one Nuo-PB and one nonNuo-PB (Fig. 5a, c). The pattern of YHB-YFP PBs is thus similar to that of phyB-CFP at 27 °C with only thermostable PBs (Fig. 3a, c). The pattern of YHB-YFP PBs stayed the same between light- and dark-grown YHB seedlings (Fig. 5a, c–e). The level of YHB-YFP increased

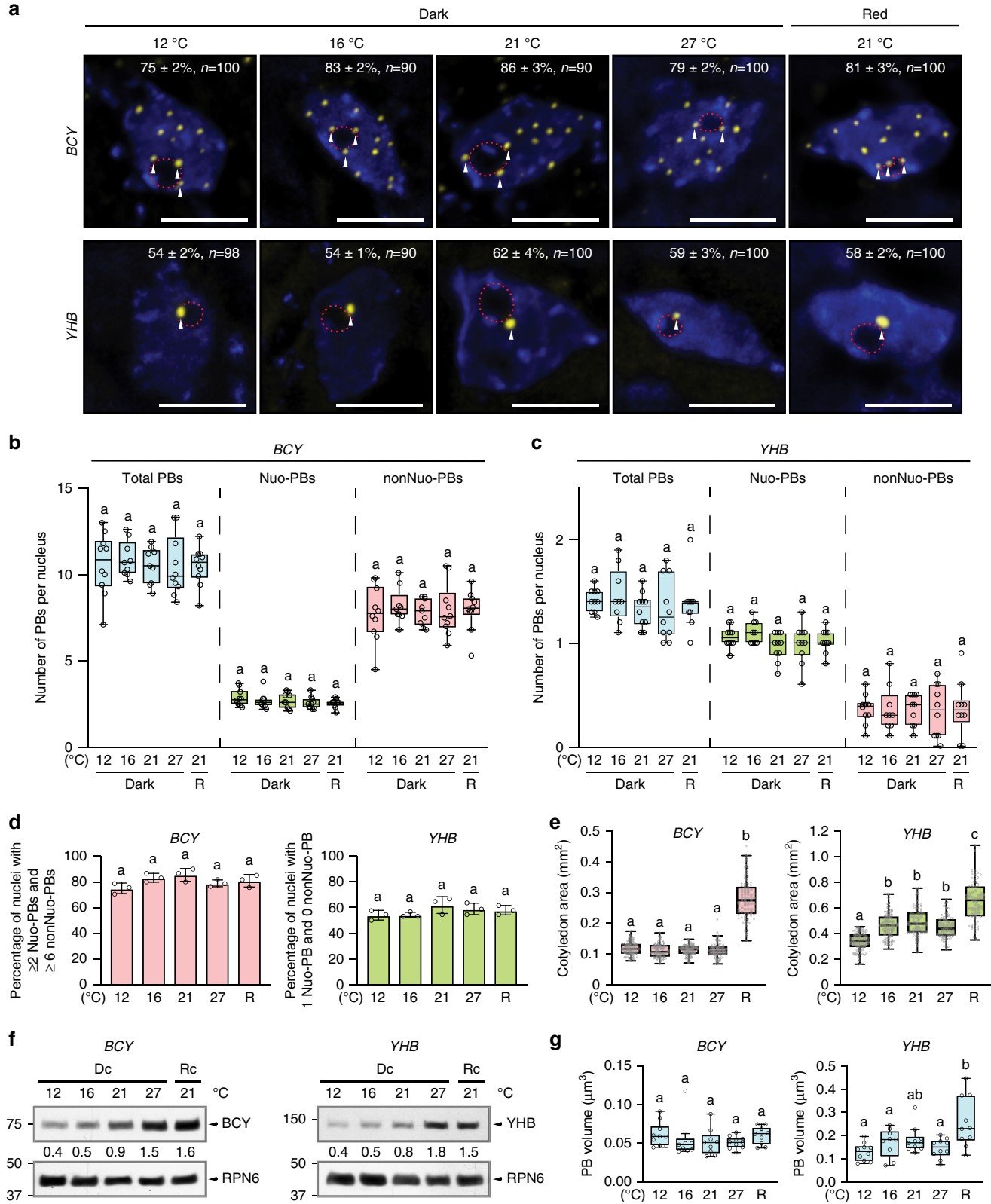

with temperature; in this case, we also observed an increase in the size of YHB-YFP PBs at 27 °C (Fig. 5f, g). Together, these results show that disruption of the ability of conformational changes within the N-terminal module of phyB abrogates the temperature responsiveness of PBs, supporting the notion that the thermosensitivity of PBs relies on the N-terminal module.

The fact that YHB localizes preferentially to only a couple of thermostable PBs as opposed to multiple PBs like phyB-CFP suggests that either the assembly of thermosensitive PBs might require the photo or thermal reversion of phyB or the specific conformation of YHB only permits its assembly to selective thermostable PBs.

**Fig. 5 BCY and YHB-YFP display distinct temperature-insensitive PB patterns. a** Maximum-projection, deconvolved images showing representative steady-state patterns of phyB C-terminal module fused with YFP (BCY) or YHB-YFP PBs in cotyledon epidermal-cell nuclei of 4-day-old *BCY* and *YHB* seedlings, respectively, grown at 12, 16, 21, and 27 °C in the dark or in 10 μmol m$^{-2}$ s$^{-1}$ in R light. Nuclei were labeled by DAPI (blue), white arrowheads indicate Nuo-PB, and the boundaries of the nucleoli are traced by red dashed lines. The percentage of nuclei with s.e. showing the representative PB pattern is shown in each image; *n* indicates the total number of nuclei analyzed. Scale bars represent 5 μm. **b, c** Quantification of the numbers of total, Nuo, and nonNuo PBs per nucleus in cotyledon epidermal-cell nuclei of the *BCY* (**b**) and *YHB* (**c**) seedlings described in **a**. **d** Quantification of the percentage of nuclei with the indicated number of PBs in cotyledon epidermal-cell nuclei of the *BCY* and *YHB* seedlings described in **a**. Error bars represent s.e. calculated from groups of seedlings. Different letters denote statistically significant differences (ANOVA, Tukey's HSD, *P* ≤ 0.05, *n* = 3). **e** Cotyledon size measurements of the *BCY* or *YHB* seedlings described in **a**. **f** Immunoblot results showing the levels of BCY or YHB-YFP in 4-day-old *BCY* or *YHB* seedlings grown in the dark or 10 μmol m$^{-2}$ s$^{-1}$ R light in the indicated temperatures. BCY and YHB-YFP were detected by anti-phyB antibodies. RPN6 was used as a loading control. The relative levels of BCY and YHB-YFP normalized against the corresponding levels of RPN6 are shown. **g** Quantification of the volume of BCY and YHB-YFP PBs in cotyledon epidermal-cell nuclei of the *BCY* and *YHB* seedlings described in **a**. For **b, c, e, g**, the boxes represent from the 25th to the 75th percentile and the bars equal the median values. Samples with different letters exhibit statistically significant differences (ANOVA, Tukey's HSD, *P* ≤ 0.05, for **b, c**, and **g**, *n* ≥ 9, for **e**, *n* ≥ 78). The source data underlying the PB measurements in **b–d, g**, cotyledon measurements in **e**, and immunoblots in **f** are provided in the Source Data file.

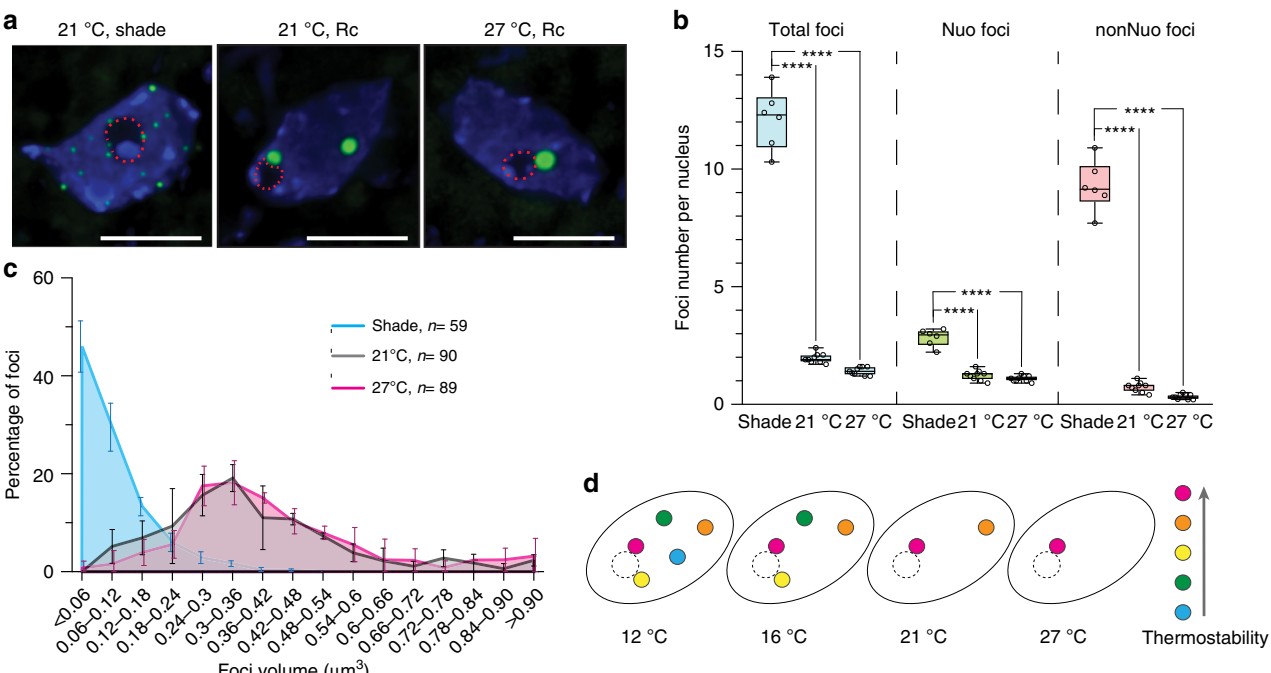

**Fig. 6 Warm temperature and shade elicit distinct photobody dynamics. a** Maximum-projection, deconvolved fluorescence microscopy images showing representative steady-state patterns of phyB-GFP in cotyledon epidermal-cell nuclei of 4-day-old *PBG* seedlings grown at 21 °C in 10 μmol m$^{-2}$ s$^{-1}$ continuous R light (21 °C, Rc), at 21 °C in 10 μmol m$^{-2}$ s$^{-1}$ R light supplemented with 10 μmol m$^{-2}$ s$^{-1}$ FR light (21 °C, shade), and at 27 °C in 10 μmol m$^{-2}$ s$^{-1}$ R light. **b** Quantification of the numbers of total, Nuo, and nonNuo PBs or foci per nucleus in cotyledon epidermal-cell nuclei of the *PBG* seedlings described in **a**. The boxes represent from the 25th to the 75th percentile and the bars equal the median values. Asterisks represent statistically significant differences in PB number (unequal variances *t* test, *P* ≤ 0.0001, *n* ≥ 6). **c** Distribution of the volume of PBs or foci in cotyledon epidermal-cell nuclei of the *PBG* seedlings grown in the indicated shade, 21 °C, and 27 °C conditions described in **a**. Error bars represent s.d. **d** Schematic illustration of the model of temperature-dependent PB dynamics. The circles filled with different colors represent PBs with a gradient of distinct thermostabilities. The circle with a dotted line represents the nucleolus. The most thermostable PB persists in all temperatures, whereas the less thermostable ones form only in lower temperatures. As a result, temperature increases progressively trigger the disassembly of individual PBs. The source data underlying the PB measurements in **b, c** are provided in the Source Data file.

**Warm temperature and shade induce distinct PB dynamics.** The current model suggests that warm temperature attenuates phyB functions by enhancing the thermal reversion of phyB to its inactive Pr form. Therefore, warm temperature signaling is thought to work in a similar manner as shade, which promotes the photoconversion of phyB to the Pr. Previous studies have shown that an increase in FR light under continuous R light or an end-of-day FR treatment leads to the relocalization of phyB-FP to many small foci and the nucleoplasm[40]. However, the temperature-dependent PB dynamics we observed are quite different from the published PB response by FR or shade. To

confirm this discrepancy between warm temperature and shade in the same cell type, we characterized the localization pattern of phyB-GFP in cotyledon epidermal-cell nuclei of *PBG* grown at 21 °C in a simulated shade condition with 10 μmol m$^{-2}$ s$^{-1}$ R light supplemented by 10 μmol m$^{-2}$ s$^{-1}$ FR light. We found that all phyB-GFP-containing PBs disappeared under the shade condition, phyB-GFP localized to many smaller subnuclear foci (Fig. 6a–c). The numbers of both Nuo and nonNuo foci increased in the shade condition compared with the PBs in 10 μmol m$^{-2}$ s$^{-1}$ R light at 21 and 27 °C (Fig. 6b). Because it is rather arbitrary to define large PBs and small foci, we looked at

the distribution of the sizes of the PBs or foci in the three datasets (Fig. 6c). The results clearly show that the transition to small foci appeared unique to the shade response, as the size distribution of PBs remained the same between 21 and 27 °C (Fig. 6c). These results, therefore, demonstrate that warm temperature and shade induce distinct PB dynamics.

## Discussion

The assembly of phyB to PBs is one of the earliest cellular events in phyB signaling[36,38], but how PB dynamics are involved in temperature sensing is less understood. Here we systematically examined the PB patterns in two *Arabidopsis* ecotypes, Col-0 and L*er*, and in two tissue/organ types, hypocotyl and cotyledon, across the ambient temperature range from 12 to 27 °C. Our results demonstrate that increasing temperature triggers a linear response in PB number in all cells examined. Surprisingly, increases in temperature from 12 to 27 °C elicit progressive disassembly of only selective temperature-sensitive or thermounstable PBs, while thermostable PBs persist even at warm temperatures, suggesting that individual PBs possess distinct thermostabilities and could serve as temperature sensors (Fig. 6d). We also defined two types of PBs, Nuo-PBs and nonNuo-PBs, and found that the dynamic changes in PB number by temperature, ecotype, and tissue/organ type are mainly contributed by changes in nonNuo-PBs, and that one of the Nuo-PBs is particularly thermostable. The thermostability of PBs relies on phyB's N-terminal photosensory module. The temperature-dependent PB dynamics reflect the enhanced thermal reversion of active phyB by warmer temperatures and correlate with the responsiveness of temperature-induced morphological changes, but unexpectedly, do not always agree with the effects of temperature on phyB signaling. For example, hypocotyl and cotyledon epidermal cells display similar temperature-dependent PB dynamics despite the opposing temperature effects on the function of phyB signaling in regulating hypocotyl and cotyledon growth, suggesting that the morphological effects of temperature could be determined by tissue/organ-specific signaling circuitry either downstream of PB dynamics or independent of phyB[18]. Together, our results unveil a unique early temperature signaling mechanism via the assembly/disassembly of individual thermosensitive PBs (Fig. 6d), which is largely different from the disappearance of all PBs induced by shade.

Our results revealed unexpectedly that individual PBs exhibit distinct thermostabilities, thereby sensitive to a specific temperature range (Fig. 6d). This conclusion is supported by the incremental decreases in PB number with temperature increases in the cotyledon and hypocotyl epidermal cells of both *PBG* and *PBC* seedlings (Figs. 2 and 3). Only a subset of PBs disappeared with temperature increases, there was at least one Nuo-PB that could persist in warm temperatures in all the cells examined (Figs. 2–4). Even thermosensitive PBs did not disappear at once (Figs. 2 and 3). PB localization of phyB has been suggested to stabilize the Pfr form or attenuate thermal reversion[39]; our results therefore suggest that individual PBs might stabilize Pfr to different extents and thereby causing the distinct temperature requirements for phyB disassembly among PBs—that is, individual PBs carry distinct thermosensitivities (Fig. 6d). This phenomenon was best exemplified in the cotyledon epidermal nuclei of *PBG*, which contained mostly two PBs—a Nuo-PB and a nonNuo-PB—allowing us to distinguish and follow the responses of particular PBs (Fig. 3a, b). Strikingly, while the Nuo-PB was insensitive to the temperature range between 12 and 27 °C, the nonNuo-PB was sensitive to and became unstable specifically at 27 °C, and disappeared rapidly during the 21 to 27 °C transition (Figs. 3a, b and 4). These results further support the conclusion

that individual PBs carry distinct thermostabilities and can respond to a specific range of temperatures, thereby implying that individual PBs could act as temperature sensors (Fig. 6d). This model does not exclude the scenario in which the assembly of phyB into PBs requires a seeding or initiating mechanism that is temperature dependent.

We also observed that the volume of PBs increased with temperature increases in both hypocotyl and cotyledon cells in *PBC* and *PBG* (Figs. 2 and 3). This is likely due to a combination of reduction of PB number and increase in phyB levels (Figs. 2 and 3). It is important to note that at any given temperature, the sizes of PBs always fall into a relatively large range with significant variations (Figs. 2e and 3f), and therefore might not be sensitive enough to report a statistically significant change in PB size. For instance, the data did not show a significant change in PB volume during the 21 to 27 °C transition in *PBG* cotyledon cells (Fig. 4e), which could be due to the lack of resolution to recognize any small differences. We have consistently observed temperature-dependent increases in the level of phyB-FP in *PBC*, *PBG*, *YHB*, and *BCY* (Figs. 2f and 5f). Because these are transgenic lines overexpressing phyB-FPs, these results might suggest that the stability of phyB is enhanced by warmer temperatures and this property relies on phyB's C-terminal module. This is consistent with the idea that warmer temperatures enhance the thermal reversion of phyB from Pfr to Pr form, which is more stable[67]. However, further investigations are required to confirm this hypothesis.

Our results show that the number of PBs varies in different tissue/organ types and *Arabidopsis* ecotypes (Figs. 2b and 3b, c). Interestingly, the variations in PB number by tissue/organ type, ecotype, and temperature are mainly contributed by nonNuo-PBs (Figs. 2b and 3b, c). Therefore, it is tempting to speculate that the presence or absence of a PB, particularly those nonNuo-PBs, could account for tissue/organ- and ecotype-specific responsiveness to temperature. This point needs to be validated by future investigations on the role of individual PBs in tissue/organ-type and ecotype-specific light and temperature responses at single-cell level. It is unclear what determines the number of PBs. One hypothesis is that PB number is controlled by the stability of PBs, alternatively, it could also be determined by a seeding mechanism for initiating PB assembly and this seedling mechanism can be influenced by ecotype-background and tissue/organ type. Our results reveal that warm temperatures impose opposing effects on the function of phyB signaling in the hypocotyl and cotyledon (Fig. 1). While it is well known that warm temperature attenuates the activity of phyB in inhibiting hypocotyl elongation, we found that it actually enhances the function of phyB signaling in promoting cotyledon expansion (Fig. 1). However, despite the opposite temperature effects, both hypocotyl and cotyledon cells showed similar PB dynamics, suggesting that the discrepancy in the temperature-dependent hypocotyl and cotyledon responses might be due to tissue/organ-specific signaling circuitry either downstream of PB dynamics or independent of phyB[18]. Consistent with this hypothesis, it has been shown that a group of auxin-responsive *SAUR* (*Small Auxin Up RNA*) genes respond differently to light signals between the hypocotyl and cotyledon[68]. It has also been shown that phyB promotes hypocotyl elongation at warm temperatures; however, our experiments were not particularly designed to test a correlation between PB dynamics and this particular response[69]. Notably, the temperature-induced PB dynamics observed in our study are different from the report by Legris et al.[18] showing that the size of PBs peaks at 20 °C; instead, we observed continuous increases in PB volume from 12 to 27 °C. These discrepancies are likely attributable to the different experimental conditions and cell types used between the two studies. While we focused on the steady-state PB patterns in

particular temperatures, in epidermal cells of the cotyledon and hypocotyl, and in seedlings grown in continuous monochromatic R light, the experiments by Legris et al.[18] were carried out during temperature transitions, in epidermal and sub-epidermal cells of the hypocotyl, and in seedlings grown in diurnal white light conditions. As our results indicated that the PB number can be influenced largely by genetic and environmental variations, it is expected that different experimental settings could result in different PB patterns. Another possible reason for different observed PB patterns by temperature may be contributed by the different sample preparation techniques used in the two studies[18]. With that said, our main conclusion that temperature increases reduce the number of PBs is consistent with the decline in the Pfr level with temperature increases shown by the previous studies[18,19].

What determines the thermostability of a PB and why individual PBs show different thermostabilities? To understand these questions, we need to examine the proposed kinetics of the photoconversion and thermal reversions between the phyB dimers of Pfr:Pfr and Pr:Pr[18,27,43]. Activation or inactivation of the phyB dimer occurs in two steps: the conversion between Pr:Pr homodimer and Pfr:Pr heterodimer and the conversion between Pfr:Pr heterodimer and Pfr:Pfr homodimer[18,43]. Both photoconversion and thermal reversions are expected to alter the PB localization of phyB. The thermal reversion from Pfr:Pfr to Pr:Pr involves two kinetic parameters: a slow Pfr:Pfr to Pfr:Pr reversion and a 100-fold faster Pfr:Pr to Pr:Pr reversion[19,43]. The current model posits that temperature influences phyB activity mainly through the conversion from Pfr:Pfr to Pfr:Pr in the dark or during nighttime, but through the conversion from Pfr:Pr to Pr:Pr in the light or during the daytime, because under illumination the Pfr:Pfr to Pfr:Pr conversion is significantly slower than the photoconversion rate from Pfr:Pr to Pfr:Pfr, and therefore can be neglected[18,19]—that is, temperature indirectly affects Pfr:Pfr stability in the light[18]. Because PB localization requires the dimeric Pfr:Pfr form of phyB[43,46], the thermostability of a PB is likely determined by the stability of the Pfr:Pfr dimers in the PB. Consistent with this notion, our results showed that the thermosensitivity of PBs is determined by the N-terminal photosensory module of phyB. Deleting (i.e., BCY) or mutating (i.e., YHB) the photosensory module resulted in temperature-insensitive PBs (Fig. 5). Moreover, increasing the intensity of light from 10 to 50 μmol m$^{-2}$ s$^{-1}$ R light, which stabilizes the Pfr: Pfr, promoted the assembly of more PBs at 27 °C (Fig. 3a–e). The fact that PBs exhibit a range of thermostabilities in ambient temperature implies that the stability of the Pfr:Pfr of phyB is different among PBs. This could be due to different phyB binding partners or slightly different active phyB forms among individual PBs. This hypothesis is supported by the PB pattern of YHB. Despite being recognized as biologically active, YHB represents a unique phyB-active form that absorbs R light or still shares some characteristics of the Pr form[53,62–64]. Interestingly, YHB-YFP did not localize to as many PBs as phyB-FP or the C-terminal module of phyB (BCY); instead, it localized to only one or two PBs that resembles the thermostable PBs of phyB-CFP at 27 °C (Fig. 5). These results support the idea that each PB might represent and could only accommodate a specific active-phyB state. The variation in thermostability or the Pfr:Pfr conformation among PBs could be due to differences in modifications of phyB itself and/or interactions with other PB constituents. For example, phosphorylation of phyB's N-terminal module can largely influence the stability and activity of the Pfr[70–72]. In particular, phosphorylation of Ser86 destabilizes the Pfr:Pfr form and attenuates PB localization[70,72]. Three phyB-interacting proteins have been reported to stabilize the Pfr form by suppressing its thermal reversion, they are ARABIDOPSIS RESPONSE REGULATOR 4 (ARR4)[73], PHOTOPERIODIC CONTROL OF HYPOCOTYL 1

(PCH1)[50–52], and PCH1-LIKE (PCHL)[52]. In particular, PCH1 and PCHL facilitate the PB localization of phyB[50–52]. Molecular genetic studies showed that PB localization of phyB requires a number of signaling components, including HMR (HEMERA)[44,65], RCB (REGULATOR OF CHLOROPLAST BIOGENESIS)[47], and NCP (NUCLEAR CONTROL OF PEP ACTIVITY)[48]. Both HMR and RCB interact directly with phyB[47,65]. However, it is still unclear if these proteins can influence phyB's thermal reversion. Interestingly, HMR and PCH1 are also required for thermomorphogenesis[50,56]. Future investigations are needed to determine whether the modifications of phyB and the interactions between phyB and its binding partners vary among individual PBs.

While warm temperature in the light affects Pfr:Pfr indirectly by accelerating the Pfr:Pr to Pr:Pr reversion, shade or FR directly photoconvert Pfr:Pfr to Pfr:Pr and Pr:Pr[18]. Therefore, it is conceivable that whereas temperature increases might not be able to disassemble certain PBs if their Pfr:Pfr is particularly stabilized, shade should be forceful enough to photoconvert phyB in all PBs. In support of this model, certain *phyB* mutants that block thermal reversion only failed to respond to temperature, but could still respond to shade[18]. Therefore, theoretically, warm temperatures could result in a PB pattern different from a shade treatment[18]. Our results demonstrate that warm temperatures and shade indeed elicit distinct PB dynamics. Temperature increases eliminate the localization of phyB-FPs to only selective thermosensitive PBs (Figs. 2 and 3), whereas all PBs disappeared in the simulated shade condition (Fig. 6a–c). It is important to point out that the PB responses shown in monochromatic R light or R light supplemented with FR light could be modulated in the natural light conditions wherein multiple classes photoreceptors are in action. Nonetheless, these results provide direct evidence that warm temperature and shade induce different PB dynamics. It is unclear how phyB-FP localization transitions from a few PBs to tens of small foci. Transitions between small foci and PBs also occur during the dark-to-light and light-to-dark transitions[36,40], suggesting that small foci might assemble into large PBs. However, it is equally possible that PBs and small foci represent discrete sites and that phyB-FPs can redistribute between these two types of sites in a light-regulated manner. Future work will focus on investigating the mechanisms underlying the distinct PB dynamics by warm temperature and shade, as well as the consequences brought about by the distinct PB patterns.

## Methods

**Plant materials and growth conditions**. The PBG[36], PBC[55], YHB[53] and BCY[55] lines of *Arabidopsis* were previously described. Seeds were surface sterilized[65] and plated on half-strength Murashige and Skoog media with Gamborg's Vitamins (MSP0506, Caisson Laboratories, North Logan, UT), 0.5 mM MES (pH 5.7), and 0.8% (w/v) agar (A038, Caisson Laboratories, North Logan, UT). Seeds were stratified in the dark at 4 °C for 5 days before treatment of specific light and temperature in an LED chamber (Percival Scientific, Perry, IA). For temperature experiments, seedlings were grown first in 21 °C for 2 days (48 h), and then either kept at 21 °C or transferred to 12, 16, or 27 °C for 2 additional days (48 h) before characterization. The light condition was maintained constant at 10 or 50 μmol m$^{-2}$ s$^{-1}$ R light for the entire 4 days. For the shade condition, seedlings were grown in a mixture of 10 μmol m$^{-2}$ s$^{-1}$ R and 10 μmol m$^{-2}$ s$^{-1}$ FR light at 21 °C for 4 days. For dark treatment, BCY and YHB seeds were exposed to 10 μmol m$^{-2}$ s$^{-1}$ FR light for 3 h immediately after stratification to deactivate the remaining phys and induce germination[65], and then kept in the dark at 21 °C for 45 h before a treatment of an indicated temperature for 2 additional days. Fluence rates of light were measured using an Apogee PS200 spectroradiometer (Apogee Instruments Inc., Logan, UT).

**Hypocotyl and cotyledon measurement**. Seedlings were mounted in phosphate-buffered saline (PBS) on a slide and then scanned using an Epson Perfection V700 photo scanner. Hypocotyl length and cotyledon size were measured using the NIH ImageJ software (http://rsb.info.nih.gov/nih-image/).

**Seedling preparation and mounting**. For steady-state PB analysis, seedlings were fixed following the protocol as described[58] with slight modifications. Seedlings were fixed under vacuum with 1% paraformaldehyde in PBS in the same growth conditions for 10 min. After quenching with 50 mM NH$_4$Cl, the fixed seedlings were permeabilized with 0.2% Triton X-100 in PBS, and nuclei were stained with 3.6 μM 4′,6-diamidino-2-phenylindole (DAPI) in PBS for 10 min. Seedlings were washed with PBS before mounted on a slide using Prolong$^{TM}$ Diamond Antifade Mountant (Thermo Fisher Scientific, Waltham, MA). The slides were left to cure overnight in the dark before being sealed with nail polish and stored at 4 °C in the dark. For all samples, we always observed both live and fixed samples, and no difference between these two conditions was observed in seedlings grown in continuous R light. However, we found that fixation weakened the signals of some very small PBs for the shade-treated *PBG* samples. Therefore, we chose to characterize the shade samples using live-cell imaging. For live-cell imaging, a seedling was mounted with PBS and transported immediately to the microscope within an aluminum foil-wrapped petri dish. Nuclei were imaged within 5 min of mounting.

**Fluorescence deconvolution microscopy and imaging analysis**. Three-dimensional image stacks of individual nuclei from cotyledon epidermal or upper hypocotyl epidermal cells were imaged using a Zeiss Axio Observer.Z1 inverted microscope equipped with a Plan-Apochromat 100×/1.4 oil-immersion objective and an Axiocam 506 mono camera (Carl Zeiss, Jena, Germany). Fluorescence was detected by using a broad spectrum X-Cite 120LED Boost high-power LED illumination system (Excelitas Technologies Corp., Waltham, MA) and the following Zeiss filter sets: DAPI, exciter 365 nm, emitter 445/50 nm/nm (Zeiss Filter Set 49); GFP, exciter 470/40 nm/nm, emitter 525/50 nm/nm (Zeiss Filter Set 38); CFP, exciter 436/25 nm/nm, emitter 480/40 nm/nm (Zeiss Filter Set 47); YFP, exciter 500/25 nm/nm, emitter 535/40 nm/nm (Zeiss Filter Set 46). Image stacks with a Z-step size of 0.25 μm were subjected to iterative classic maximum likelihood estimation deconvolution using Huygens Essential (Scientific Volume Imaging, Netherlands) with the following parameters: 50 iterations, signal-to-noise ratio 40, and quality threshold 0.01. The object analyzer tool was used to threshold the image and to calculate the number and volume of the PBs. For each nucleus, the distinction between a Nuo-PB and a nonNuo-PB was manually performed. The maximum projection of the deconvolved sections was exported to TIFF (Tagged Image File Format) and then processed by Adobe Photoshop CC (Adobe, San Jose, CA).

**Protein extraction and immunoblots**. Total protein was extracted from *Arabidopsis* seedlings grown under the indicated conditions. Plant tissues were ground using a Mini-Beadbeater-24 (BioSpec Products Inc.) in three volumes (mg/μL) of extraction buffer containing 100 mM Tris-Cl, pH 7.5, 100 mM NaCl, 5 mM EDTA, 5% sodium dodecyl sulfate (SDS), 20 mM dithiothreitol, 40 mM β-mercaptoethanol, 2 mM phenylmethylsulfonyl fluoride, 80 μM MG115 (Sigma-Aldrich), 80 μM MG132 (Sigma-Aldrich), 1× EDTA-free protease inhibitor cocktail (Roche), 20% glycerol, 10 mM *N*-ethylmaleimide and 0.01% bromophenol blue and immediately boiled for 10 min, followed by centrifugation at 16,000 × g for 10 min. Proteins in the supernatant were separated by SDS–polyacrylamide gel electrophoresis and transferred to nitrocellulose membranes for immunoblotting. Primary antibodies, including monoclonal mouse anti-phyB antibodies (gift from Akira Nagatani) and polyclonal rabbit anti-RPN6 antibodies (Enzo Life Sciences, BML-PW8370-0100) were used at 1:1000 dilution. Goat anti-mouse (Bio-Rad, #1706516) and goat anti-rabbit (Bio-Rad, #1706515) secondary antibodies were used at a 1:5000 dilution. Signals were detected by chemiluminescence using a SuperSignal kit (Thermo Fisher Scientific) and films (Phenix Research Products), and the bands were quantified by using the NIH ImageJ software. The relative levels of phyB-FPs were normalized against the corresponding levels of RPN6.

**Reporting summary**. Further information on research design is available in the Nature Research Reporting Summary linked to this article.

## Data availability
The source data underlying Figs. 1a–d, 2b–f, 3b–f, 4b–e, 5b–g and 6b, c are provided as a Source Data file.

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

## Acknowledgements

We thank Ruth Jean Ae Kim, Chan Yul Yoo, Soeun Han, and Isaac Diaz for their critical comments and suggestions regarding the manuscript. This work was supported by Grant 2R01GM087388 from the National Institute of General Medical Sciences to M.C.

## Author contributions

J.H., K.K., and M.C. conceived the original research plan; J.H. and K.K. characterized the PB morphology and thermomorphogenic responses; Y.Q. and K.K. performed the immunoblot experiments; J.H., K.K., Y.Q., and M.C. analyzed the data; M.C., J.H., and K.K. wrote the article with contributions from all authors.

## Competing interests

The authors declare no competing interests.
