## [Peer Review File · Nature Communications]

Reviewers' comments:

Reviewer #1 (Remarks to the Author):

Photoactivated phyB forms subnuclear structures, photobodies, in the nucleus of plant cells and this process represents one of the earliest events in light signaling. Although the precise function of these PBs in light signaling is still under debate, accumulating evidence has revealed that PBs could serve as storage sites for activated phyB and possibly signaling intermediates that interact with the active phyB form. It was shown that thermal reversion of phyB Pfr is inhibited in PBs resulting in stabilization of phyB Pfr and maintenance of phyB signaling after light-dark transfer. Thermal reversion of phyB is strongly temperature dependent which represents the underlying mechanism for phyBs ability to function as temperature sensor. Previous studies have already shown that PBs respond to temperature and shade.

In the manuscript by Hahm et al. the authors aimed to investigate how phyB containing PBs respond to temperature. They analyzed PB steady state patterns in epidermal cells of hypocotyl and cotyledons in two different Arabidopsis ecotypes grown in red light at different temperatures and found that PB number decreased with increasing temperature. In a kinetic analysis they further show that temperature dependent PBs disappear within 6 h after transfer from 21 to 27 °C. Interestingly they found that PBs in proximity to the nucleolus were less responsive to temperature compared to other PBs in the nucleus. They further show that PBs containing only the C-terminal half of phyB without the photosensory domain are temperature insensitive and that temperature and simulated shade induce different PB patterns. Without additional evidence, the authors propose a model in which kr_2 , the thermal reversion rate of the Pfr:Pfr homodimer, is the determinant for PB stability and that individual PBs could have different kr_2 explaining their different thermal stabilities.

Although the authors raise potentially interesting questions about temperature dependent PB dynamics and the experimental data are sound, the study remains predominantly descriptive with only weak originality at the mechanistic level. The interesting new observations that PB patterns differ between ecotypes and that PBs within one nucleus respond differently to temperature are not investigated mechanistically and the explanations provided by the authors are speculative. In addition I really have problems with the proposed hypothetical model. It is not supported by experimental or theoretical data and in my eyes, the authors have disregarded important features of phyB dynamics that would be incompatible with such model. Hence I cannot recommend the manuscript in its present form for publication in Nature Communications.

Please find my comments below:

(1) PBs are dynamic structures, phytochrome molecules are constantly binding and unbinding with very fast rates (Rausenberger et al., 2010). Importantly, it was demonstrated that thermal reversion takes place in the nucleoplasm and is suppressed in the PBs (Klose et al. 2015). The Chen lab even has shown earlier that phyB Pfr must be stabilized in PBs (Van Buskirk et al., 2014). The 3-state-model of the phyB dimer that the authors utilize for their model is only valid under conditions not inducing formation of the late-type PBs (Klose et al., 2015, Legris et al., 2016). Further, the experiments described in the study are performed in continuous red light ($10 \mu\text{mol m}^{-2} \text{s}^{-1}$), where Pr to Pfr photoconversion is more than two orders of magnitude faster than kr_2 , even at 27 °C. Therefore it is very unlikely that kr_2 determines the thermostability of the PBs or how fast phyB dissociates from PBs.

PhyB Pfr dissociates rapidly from PBs allowing thermal reversion in the nucleoplasm. In Pfr:Pfr it has a high chance to rebind to a PB. At higher temperature, when kr_1 (and kr_2) is increased, it simply takes longer until a dimer reaches the Pfr:Pfr state and this could affect the equilibrium between nucleoplasm and PB. In an alternative scenario, the individual PBs could have different binding affinities for phyB, possibly because of different protein composition, and this could lead to a redistribution between them where the "higher affinity" PB wins.

Have the authors tested PB temperature dependence at higher fluence rates of red light to see

whether increasing light intensity can compensate the high temperature effect?

(2) Under the light conditions used ($10 \mu\text{mol m}^{-2} \text{s}^{-1}$) I would expect minor effects of increased temperature on the overall Pfr photoequilibrium. Therefore it would be interesting to see whether the alterations in PB patterns are physiologically relevant? Have the authors tested some phyB responses?

(3) I find it difficult to conclude from the steady state PB analyses shown in Fig. 1-2 that individual PBs "vary in thermostability - i.e., the temperature requirement for triggering the disassembly of phyB-GFP from PB varies among individual PBs" (line145). It is possible that some PBs simply do not develop under higher temperature conditions, from just looking at the steady state pattern it is not possible to conclude about disassembly.

(4) Further, since the authors claim that there is a redistribution of phyB to fewer PB numbers coinciding with enhanced phyB accumulation I wonder why they have not investigated whether phyB redistributes also to the nucleoplasm at higher temperatures (can be done by quantifying fluorescence intensity outside the PBs). This would be particularly interesting after transfer from 21 to 27 °C, since here the remaining PBs do not increase in size and phyB levels are stable.

(5) I do not understand why there is no change in PB size observed in Fig. 3E. I would expect that the thermosensitive PB shrinks before it disappears. Further, images chosen for Fig. 3A suggest that PBs increase in size.

(6) Any idea why PB number is variable between ecotypes? Other proteins seems to be important for PB formation, PCH1 ox for example has much higher PB number.

(7) The authors might consider to use "thermal reversion" rather than "dark reversion" as it is preferred based on current state of research.

(8) Why are the numbers of replicates so low (e.g. $n=3/$ in Fig. 1C/D) when the authors have 90-100 images?

Reviewer #2 (Remarks to the Author):

The manuscript submitted by Hahm et al. describes photobody (PB) dynamics (such as steady state patterns, PB numbers and sizes, and PB locations in the nucleus) of phytochrome B (phyB) in a temperature-dependent manner. Specifically, the authors investigated the PB dynamics of phyB (with GFP or CFP fusion) under different temperatures (12, 16, 21, and 27°C), with two ecotypes (Ler and Col-0), and also with two epidermal cells (hypocotyl and cotyledon). Moreover, a YFP-fused C-terminal domain of phyB (BCY-YFP) and a YFP-fused constitutively active mutant of phyB (YHB-YFP) were included in their analysis to support their model. Lastly, they compared the PB dynamics between a simulated shade condition (R+FR, each $10 \mu\text{mol}\cdot\text{m}^{-2}\cdot\text{s}^{-1}$) and a warm temperature condition (27°C). Their results suggest that the PB dynamics are different between Ler and Col-0 ecotypes and also between hypocotyl and cotyledon cells, suggesting potentially diverse mechanisms in ecotype- and cell-specific thermal sensitivities. More importantly, they found that the PB dynamics are also different between shade and high temperature conditions, in which the transition from large to small PBs induced by shade (i.e., weak light or FR) conditions was not observed in the warm temperature condition. Results showed that progressive increases in temperature reduced the number of PBs, but enlarged the size of PBs. Here, the authors divided PBs into two types, nucleolar-associated photobody (Nuo-PB) and non-nucleolar-associated photobody (nonNuo-PB), and also found different thermal sensitivities between Nuo-PBs and nonNuo-PBs.

Overall, this study may enlarge our understandings the PB dynamics of phyB, which is helpful to elucidate the function of phyB in plants. The contents and organization of the manuscript are easy to follow the work done for this study. However, here are some comments to improve the manuscript.

Major comments:

1. The data in this manuscript are mostly about the analysis of phyB images in the nucleus (such as PB patterns, including numbers, sizes, and association with the nucleolus), so the readers might not understand the significance of these data. Thus, the authors need to show the seedling

phenotypes under the temperature conditions that the PBs were analyzed in this study. These are important to recognize different thermo-responses between Ler and Col-0 ecotypes (and also those of BCY and YHB). In addition, the cell patterns (sizes and numbers) in hypocotyls and cotyledons need to be compared under different temperature conditions used in this study.

2. The important data of this study would be the PB dynamics of phyB under warm (or high) temperatures. However, this study used only one high temperature condition (27°C). If we consider the optimum temperature for the growth of Arabidopsis plants (23-25°C in Methods used by the Arabidopsis Biological Resource Center), it would be necessary to use at least two high temperature conditions to support the model suggested in this study. Probably, 30°C may be used, because this temperature was used in other references (such as Legris et al., 2016).

3. The PB images of this study were obtained cells from fixed seedlings (with the method described in their recent paper; Yoo et al., 2019), so it looks like that the PB patterns are different from live cells in other references. Thus, it would be better to include PB images from living seedlings (probably as supplementary figures). These data may also help readers for understanding the importance of PB formation in plant thermomorphogenesis. As another question, is there any possibility to lose small PBs during the fixation steps? [consider the data differences in other references such as Legris et al., 2016 (reduced PBs in both numbers and sizes) and Huang et al., 2019 (more PBs of YHB than this study, even in the same R light condition (1-2 vs. 4-5))]

4. An interesting result of this study is that phyB protein levels (including BCY and YHB) are quite increased in warm temperature condition (27°C). Is there any answer to explain these data? Is phyB degradation delayed in this condition? Isn't the increased level related to phyB function? Considering that the western data do not distinguish between Pr and Pfr forms, and also different locations (cytoplasm vs. nuclear), the authors need to perform more western blots with cytosolic and nuclear fractions. In addition, if possible, they need to analyze the spectra of phyB to confirm the Pr and Pfr forms (or ratios).

5. In this paper, the authors divided PBs into Nuo-PBs and nonNuo-PBs, and showed that the temperature-dependent variations in PB numbers are contributed mainly by nonNuo-PBs. Then, are the formation mechanisms (and/or components) different between Nuo-PBs and nonNuo-PBs? (For example, Nuo-PBs function as basic roles of phyB, and nonNuo-PBs function as transient roles of phyB depending on fluctuating environmental conditions such as light and temperature) For this, the authors may try to investigate the function of Nuo-PBs, probably through gene expression analysis using the conditions in the presence and absence of nonNuo-PBs. Thus, please provide any evidence for the different functions between Nuo-PBs and nonNuo-PBs.

Minor comments:

- It would be better to indicate Nuo-PBs in Figure 4.
- In the discussion (lines 390-394), the authors suggest the importance of phyB phosphorylation/modifications in the PB formation. Then, please describe how phyB can be phosphorylated/modified in plants (probably by phyB's intrinsic kinase activity or other kinases).
- This study used a constitutively active mutant of phyB, named YHB. Recently, there is another report of constitutively active phyB mutant (named AtYVB in Jeong et al., *Plant Physiol.*, 171: 2826-2840). Is there any difference between YHB and AtYVB (in terms of phyB function and nuclear localization)? If the two mutants are different, it would be interesting to compare the PB formation with YHB and YVB.
- Check typo-errors in the manuscript. For example, line 121 ("four"), line 376 ("Tte"), and line 373 ("thermostabilitis"), and so on.
- Check also reference format (especially titles, some have capital letters in each word).

Reviewer #3 (Remarks to the Author):

The work by Hahm et al investigates the effect of temperature on the sub-nuclear localization of the light and temperature receptor phytochrome B (phyB). Using fluorescence microscopy, they describe the abundance and size distribution of phyB-containing nuclear bodies (photobodies).

Although it had already been described that temperature has an effect on the size distribution of photobodies in plants grown in white light (Legris et al Science 2016), this paper shows that when plants grow in constant red light the temperature effects on photobodies is different from the previously reported temperature effects on photobodies in white light. Under continuous red light illumination, higher temperatures, which decrease the proportion of phyB in its active form, reduce the abundance of large photobodies while increasing their volume. This occurs both in cotyledons and hypocotyls and in two ecotypes, namely Landsberg erecta and Columbia. The authors notice that in many nuclei some photobodies are closer to the nucleolus and these frequently remain present in conditions where the average number of photobodies decreases. Shade also decreases the proportion of phyB in its active form but has a different effect on the size distribution of photobodies, which has been described previously (Trupkin et al., Plant Physiol 2014), and the authors confirm this finding and further analyze this phenomena in terms of association with the nucleolus. The main novelty of the work is the identification of a subset of photobodies that have higher stability in high temperatures and are associated to the nucleolus. I have the following suggestions/comments.

- 1) the only distinctive feature of temperature-stable photobodies is that they appear near the nucleolus in deconvoluted epifluorescence images of DAPI-stained cells. It is not shown whether they are actually associated with the nucleolus in a more direct way. The chance of finding a photobody near the nucleolus in a picture could be affected, for instance, by the size of the nucleolus (if it was larger in high temperature the chance of finding a photobody near it would be higher). Thus, I suggest that the authors measure the size of the nucleolus in the different temperature conditions to evaluate this possibility. Also the work could be improved including a nucleolar marker in the images and doing 3D reconstitutions of the nuclei to improve the description of the association between photobodies and nucleoli. The biological significance of the association to the nucleolus should be discussed.
- 2) The response of photobodies to temperature is not the same as to shade. Although both responses had been evaluated independently (Legris et al 2016, Trupkin et al 2014) no direct comparison had been done. The authors point out that in their conditions in response to high temperature they don't detect the appearance of small photobodies, typically found in response to shade. However, as far as I understood from the methods section, the procedure to evaluate each condition was different. The authors note that fixation weakened the signal of very small photobodies and used fixed samples to evaluate temperature effects but unfixed samples to evaluate shade effects. If this is the case, then the comparison is not good, and at least for this figure the high temperature experiment should be repeated without fixation to allow a fair comparison with the effect of shade. Also, to make a better comparison milder FR treatments should be tested, to reach similar Pfr levels to those obtained in continuous RL 10uE at 27°C (so far the only condition tested is 10uE RL vs 10uE RL + 10 uE FR which is expected to give a largely different PfrB/Ptot than 10 uE at 27°C). Also, in fig 5b the number of nucleolar associated and not associated photobodies is compared between shade, 21°C and 27°C. It should be noted that in the case of shade there are small and large photobodies mixed and given that the dynamics of these two types of photobodies is different probably they represent phyB in different states and thus they should be treated/analyzed separately. Could the authors at least comment on this?
- 3) Another finding is that the abundance of photobodies and their response to temperature differs between cotyledons and hypocotyl and the number differs between the accessions Columbia and Landsberg erecta. For the first comparison, can the authors hypothesize whether this has any biological significance? Could it be merely due to differences in phyB protein levels among these cells? The comparison between accessions is a bit more difficult to interpret. One main concern I have is that the transgenes used are different among accessions. Not only the protein levels are different but also the fluorophores. I suggest to test whether the reported differences remain true using other transgenic lines of the same accession, ideally with different protein levels and fluorophores. Again, it would be good to test whether this difference is biologically significant. In this case I suggest to test it by evaluating hypocotyl elongation in response to temperature in the selected wild type and transgenic genotypes.
- 4) In the discussion the authors propose that phyB localized in photobodies associated to the

nucleolus have a different kr2 than phyB in other photobodies. Based on the presented data this is a speculation and thus I think it shouldn't be in the figure. If the authors would like to test this hypothesis they could evaluate photobody dynamics using phyB mutated versions with normal photoconversion and different thermal stability (Zhang et al Plant physiol 2013, Burgie et al PNAS 2014) . An alternative to this model would be that the rate of phyB association to these nucleolar associated photobodies is increased by temperature (Rausenberger Plos One 2010 PMID: 20502669). To test this, it would be interesting to dissect the data of photobody volume into nucleolar associated and not associated photobodies (maybe it's not just that they remain but actually more phyB is being actively incorporated to the nucleolar associated photobodies in high temperatures). It should be discussed that it has been suggested that localization to photobodies inhibits dark reversion (Huang PNAS 2019 PMID: 30948632, Rausenberger Plos One 2010 PMID: 20502669, Klose New Phytol 2015 PMID: 260422).

5) It should also be discussed that in the conditions tested, i.e. continuous red light and high temperature, hypocotyl length of Columbia and Landsberg erecta is regulated by a complex mechanism where increasing light intensities first inhibit elongation but then promote it (Johansson Nat commun 2014 PMID: 25258215). Although I understand that the aim of this paper is not to explain that process, it should be noted that in these conditions more phyB in the active conformation is not more efficiently promoting photomorphogenesis, and thus the biological significance of the abundance of photobodies is difficult to predict. Overall, the work could be improved by a description of a phyB -regulated developmental response (e.g. hypocotyl elongation) in the selected light and temperature conditions where the pattern of photobodies are described in order to get a better sense of the correlation between photobodies and a developmental response.

6) Some final comments about the presentation of the results, in Fig 1c, 1d, 2d, 2e, 3b, 3c, 4d, the number of photobodies per cell is different (sometimes 1, sometimes 2). It would be easier to compare among conditions if all were representing the same values. Figure 2d and 2e show PBC and PBG in a different order from the rest of the figures and is confusing while comparing among figures.

Response to Reviewers

Reviewer #1

1. PBs are dynamic structures, phytochrome molecules are constantly binding and unbinding with very fast rates (Rausenberger et al., 2010). Importantly, it was demonstrated that thermal reversion takes place in the nucleoplasm and is suppressed in the PBs (Klose et al. 2015). The Chen lab even has shown earlier that phyB Pfr must be stabilized in PBs (Van Buskirk et al., 2014). The 3-state-model of the phyB dimer that the authors utilize for their model is only valid under conditions not inducing formation of the late-type PBs (Klose et al., 2015, Legris et al., 2016). Further, the experiments described in the study are performed in continuous red light ($10 \mu\text{mol m}^{-2} \text{s}^{-1}$), where Pr to Pfr photoconversion is more than two orders of magnitude faster than kr2, even at 27 °C. Therefore it is very unlikely that kr2 determines the thermostability of the PBs or how fast phyB dissociates from PBs. PhyB Pfr dissociates rapidly from PBs allowing thermal reversion in the nucleoplasm. In Pfr:Pfr it has a high chance to rebind to a PB. At higher temperature, when kr1 (and kr2) is increased, it simply takes longer until a dimer reaches the Pfr:Pfr state and this could affect the equilibrium between nucleoplasm and PB. In an alternative scenario, the individual PBs could have different binding affinities for phyB, possibly because of different protein composition, and this could lead to a redistribution between them where the “higher affinity” PB wins.

Have the authors tested PB temperature dependence at higher fluence rates of red light to see whether increasing light intensity can compensate the high temperature effect?

Response:

We appreciate the reviewer for clarifying the model proposed by the previous publications. We agree that it might be premature to pinpoint that the thermostability of a PB is determined specifically by kr2, because other possibilities exist as suggested by this reviewer. Therefore, we modified the model by focusing on the main conclusion that individual PBs carry distinct thermostabilities (Fig. 6d). We have also revised the corresponding section in the Discussion, highlighting different possibilities causing the distinct thermostabilities among PBs. The reviewer also suggested an interesting experiment to test whether higher light intensity could compensate the warm-temperature effect. We performed the experiments by examining the patterns of phyB-FP in the cotyledon nuclei of *PBG* and *PBC* grown in $50 \mu\text{mol m}^{-2} \text{s}^{-1}$ of R light at 27°C. The new data are included in Fig. 3. In both *PBG* and *PBC*, phyB-FP localized to more PBs in the higher intensity of light, indicating that increasing in light intensity can compensate the high temperature effect. These results thus support our conclusion that the thermostability of PBs depends on the stability of the Pfr form of phyB.

2. Under the light conditions used ($10 \mu\text{mol m}^{-2} \text{s}^{-1}$) I would expect minor effects of increased temperature on the overall Pfr photoequilibrium. Therefore it would be interesting to see whether the alterations in PB patterns are physiologically relevant? Have the authors tested some phyB responses?

Response:

We agree that it is important to examine whether the *PBG* and *PBC* lines retain similar temperature responses as their respective ecotype-background lines. Therefore, we examined temperature-induced responses in the hypocotyl and cotyledon in Col-0, *PBC* (in Col-0 background), *Ler*, and *PBG* (in *Ler* background). These results are included in the new Fig. 1. Both Col-0 and *Ler* showed significant hypocotyl elongation with temperature increases in 10 $\mu\text{mol m}^{-2} \text{s}^{-1}$ of R light (Fig. 1a,b). These results are consistent with our recent published studies on daytime phyB temperature signaling¹. *PBC* and *PBG* exhibited similar but reduced hypocotyl elongation responses likely due to the overexpression of phyB-FPs (Fig. 1). These results are consistent with the notion that warmer temperatures can attenuate the function of phyB signaling in inhibiting hypocotyl elongation. In contrast to inhibiting hypocotyl growth, phyB promotes the expansion of the cotyledon. How temperature influences the phyB functions in cotyledons has not been explicitly described previously. Our results show that increases in temperature surprisingly enhance the functions of phyB signaling in promoting cotyledon expansion in Col-0 and *Ler* in permissive high light conditions (50 $\mu\text{mol m}^{-2} \text{s}^{-1}$ R light) as well as in *PBC* and *PBG* lines in 10 $\mu\text{mol m}^{-2} \text{s}^{-1}$ R light (Fig. 1). These results indicate the *PBC* and *PBG* lines also exhibit the same temperature-dependent cotyledon responses as Col-0 and *Ler*, respectively. In contrast, the cotyledons of *YHB* and *BCY* did not respond to temperature as *PBC* and *PBG* (Fig. 5e). The new data on morphological responses, combined with the PB dynamics data, suggest that the PB dynamics correlate with the responsiveness of hypocotyl and cotyledon to temperature. More interestingly, our results show that hypocotyl and cotyledon cells displayed similar temperature-dependent PB dynamics despite the opposing temperature effects on phyB signaling, indicative of tissue/organ-specific phyB signaling circuitry downstream of PB dynamics.

3. I find it difficult to conclude from the steady state PB analyses shown in Fig. 1-2 that individual PBs “vary in thermostability - i.e., the temperature requirement for triggering the disassembly of phyB-GFP from PB varies among individual PBs” (line 145). It is possible that some PBs simply do not develop under higher temperature conditions, from just looking at the steady state pattern it is not possible to conclude about disassembly.

Response:

The conclusion of individual PBs with distinct thermostability is supported by several lines of evidence. Besides the incremental decline of PB number with temperature increases (Fig. 2 and 3), this conclusion is strongly supported by the observations of the two PBs in the epidermal nuclei in *PBG*, where we could distinguish and follow individual PBs, a Nuo-PB and a nonNuo-PB (Fig. 4). Our results show that the nonNuo-PBs were sensitive specifically to 27°C and disappeared in most cells within 6 h after a transition from 21°C to 27°C, while the Nuo-PBs were stable in all temperatures (Fig. 3 and 4). These results indicate that individual PBs have distinct thermosensitivity and thermostability. We do agree that the thermostability model does not exclude the scenario in which the assembly of phyB into PBs requires a seeding/initiating

mechanism that could be temperature-dependent. We therefore explicitly stated this possibility by adding a sentence at the end of the second paragraph of the Discussion.

4. Further, since the authors claim that there is a redistribution of phyB to fewer PB numbers coinciding with enhanced phyB accumulation I wonder why they have not investigated whether phyB redistributes also to the nucleoplasm at higher temperatures (can be done by quantifying fluorescence intensity outside the PBs). This would be particularly interesting after transfer from 21 to 27 °C, since here the remaining PBs do not increase in size and phyB levels are stable.

Response:

We agree that it would be interesting to also measure the amount of phyB-FPs in the nucleoplasm. However, because nuclei vary in volume, fluorescence signals per unit area or unit volume are not a good estimate of the nucleoplasmic signals, and therefore, it requires three-dimensional models of the nucleus for measure nucleoplasmic phyB. This is quite difficult to do. More importantly, in our opinion, these data are not absolutely necessary to draw our main conclusions that the number of PBs are influenced by temperature and individual PBs respond to temperature differently. Regardingly the changes in PB sizes, we stated in the Results that “the increases in PB size with temperature could be due to the redistribution of phyB-GFP to fewer PBs and/or the enhanced phyB accumulation.”

We added the following in the Discussion: “It is important to note that at any given temperature, the sizes of PBs always fall into a relatively large range with significant variations (Fig. 2e and 3f), and therefore, might not be sensitive enough to report a statistically significant change in PB size. For instance, the data did not show a significant change in PB volume during the 21°C-to-27°C transition in *PBG* cotyledon cells (Fig. 4e), which could be due to the lack of resolution to recognize any small differences.”

5. I do not understand why there is no change in PB size observed in Fig. 3E. I would expect that the thermosensitive PB shrinks before it disappears. Further, images chosen for Fig. 3A suggest that PBs increase in size.

Response:

Please see our response above.

6. Any idea why PB number is variable between ecotypes? Other proteins seems to be important for PB formation, PCH1 ox for example has much higher PB number.

Response:

We added the following in the Discussion: “It is unclear what determines the number of PBs. One hypothesis is that PB number is controlled by the stability of PBs, alternatively, it could also be determined by a seeding mechanism for initiating PB assembly and this seedling mechanism can be influenced by ecotype background and tissue/organ type.” We did not want to speculate

on the PCH1ox data, because PBs were qualified in different light, cell-type, and genetic backgrounds.

7. *The authors might consider to use “thermal reversion” rather than “dark reversion” as it is preferred based on current state of research.*

Response: We have changed “dark reversion” to “thermal reversion”.

8. *Why are the numbers of replicates so low (e.g. n=3/ in Fig. 1C/D) when the authors have 90-100 images?*

Response:

The error bars represent s.e. of nuclei from groups of seedlings. 90-100 are the total nuclei analyzed. We imaged 5-10 nuclei per seedling, analyzed several seedlings, and divided the nuclei into 3 groups for each data point of this analysis.

Reviewer #2

Major comments:

1. *The data in this manuscript are mostly about the analysis of phyB images in the nucleus (such as PB patterns, including numbers, sizes, and association with the nucleolus), so the readers might not understand the significance of these data. Thus, the authors need to show the seedling phenotypes under the temperature conditions that the PBs were analyzed in this study. These are important to recognize different thermo-responses between Ler and Col-0 ecotypes (and also those of BCY and YHB). In addition, the cell patterns (sizes and numbers) in hypocotyls and cotyledons need to be compared under different temperature conditions used in this study.*

Response:

We thank this reviewer for the comments. We have characterized temperature-dependent hypocotyl elongation and cotyledon expansion in *Ler*, *Col-0*, *PBG* and *PBC*. The results are included in the new Fig. 1. The cotyledon expansion data for *YHB* and *BCY* are included in Fig. 5e. We used hypocotyl length and cotyledon area as readouts, as opposed to cell size and number, because the former responses are well defined by previous studies. Regarding the temperature-dependent hypocotyl and cotyledon responses, please see our response to Reviewer 1 Question #2.

2. *The important data of this study would be the PB dynamics of phyB under warm (or high) temperatures. However, this study used only one high temperature condition (27°C). If we consider the optimum temperature for the growth of Arabidopsis plants (23-25°C in Methods used by the Arabidopsis Biological Resource Center), it would be necessary to use at least two high temperature conditions to support the model suggested in this study. Probably, 30°C may be used, because this temperature was used in other references (such as Legris et al., 2016).*

Response:

The goal of this study was not to focus on warm temperatures per se. In our opinion, the value of this study is the systematic characterization of the PB dynamics in two tissue/organ types across the ambient temperature range from 12°C to 27°C. This particular temperature range has been considered as the “ambient temperatures” for Arabidopsis because, as shown previously, this temperature range does not induce heat stress²⁻⁴. The current experimental settings allowed us to draw the main conclusion on the temperature-dependent PB dynamics in ambient temperatures. We agree that it would be interesting to expand this investigation to include cold and hot temperatures. But, as we can see, detailed imaging analysis of the PB patterns is quite time consuming. Therefore, we thought that we should focus on our efforts on ambient temperatures first and investigate hot and cold temperatures in the future.

3. The PB images of this study were obtained cells from fixed seedlings (with the method described in their recent paper; Yoo et al., 2019), so it looks like that the PB patterns are different from live cells in other references. Thus, it would be better to include PB images from living seedlings (probably as supplementary figures). These data may also help readers for understanding the importance of PB formation in plant thermomorphogenesis. As another question, is there any possibility to lose small PBs during the fixation steps? [consider the data differences in other references such as Legris et al., 2016 (reduced PBs in both numbers and sizes) and Huang et al., 2019 (more PBs of YHB than this study, even in the same R light condition (1-2 vs. 4-5))]

Response:

We have always carefully compared live-cell and fixed-cell first to determine whether the fixed-cell data are consistent with the live-cell data. Under most conditions, we do not see any difference in the PB patterns between live and fixed cells. Because, during live-cell imaging, the phyB-FP pattern could change by the excitation light (as described in Yoo et al.⁵), fixation offers a better way for accuracy and consistency. However, under shade or very dim light, some very tiny foci in the fixed samples could become very sensitive to photobleaching. So, we specifically chose live-cell imaging for these samples.

The PB patterns in the hypocotyl shown here are the same as what we have reported before⁶⁻⁹. With that said, this was the first time that the number and volume of PBs were systemically calculated using a 3D model. The PB patterns in cotyledon have not been characterized previously. The PB patterns in Legris 2016 and Huang 2019 were characterized in different conditions (white light and diurnal conditions), in different cell types, and using different imaging analysis tools, so it would be difficult to compare these results.

4. An interesting result of this study is that phyB protein levels (including BCY and YHB) are quite increased in warm temperature condition (27°C). Is there any answer to explain these data? Is phyB degradation delayed in this condition? Isn't the increased level related to phyB function? Considering that the western data do not distinguish between Pr and Pfr forms, and also different locations (cytoplasm vs. nuclear), the authors need to perform more western blots with cytosolic and nuclear fractions. In addition, if possible, they need to analyze the spectra of phyB to confirm the Pr and Pfr forms (or ratios).

Response:

We appreciate the suggestions of the reviewer. We also found the temperature-dependent increases in the phyB levels interesting, but since this is not the main point of this study, we decided to look into the details of this regulation in a separate study. A potential increase in phyB stability at warm temperatures is consistent with the idea that elevated temperatures promote the conversion to the Pr form, which is more stable. With that said, the levels of YHB and BCY also increased with temperature, suggesting that additional mechanisms are involved.

5. In this paper, the authors divided PBs into Nuo-PBs and nonNuo-PBs, and showed that the temperature-dependent variations in PB numbers are contributed mainly by nonNuo-PBs. Then, are the formation mechanisms (and/or components) different between Nuo-PBs and nonNuo-PBs? (For example, Nuo-PBs function as basic roles of phyB, and nonNuo-PBs function as transient roles of phyB depending on fluctuating environmental conditions such as light and temperature) For this, the authors may try to investigate the function of Nuo-PBs, probably through gene expression analysis using the conditions in the presence and absence of nonNuo-PBs. Thus, please provide any evidence for the different functions between Nuo-PBs and nonNuo-PBs.

Response:

We first decided to divide PBs into Nuo- and nonNuo-PBs because we realized (during the study) that these two classes of PBs showed different thermostabilities (in particular in the cotyledon cells in *PBG*). This new classification led to the discovery that the variations in PB number were largely contributed by the nonNuo-PBs. To our knowledge, this is the first study suggesting distinct roles or functions of individual PBs. As the reviewer suggests, this study raised a number of new interesting questions, such as what determines the formation of Nuo-PBs and nonNuo-PBs and whether they play different roles in the regulation of PIFs and phyB-dependent genes. We think that this study opens a new avenue to begin to address those cell biology questions in phyB signaling. However, because the PB dynamics are characterized at the single-cell level, to investigate their functions in phyB signaling, it also requires the development of single-cell techniques to examine the downstream signaling events. We are currently developing fluorescence-labeling methods particularly RNA and DNA fluorescence in situ hybridization tools to examine the function of individual PBs. As we just began these new investigations, it will take a while to figure out the roles of individual PBs.

Minor comments:

- It would be better to indicate Nuo-PBs in Figure 4.

Response:

We have labeled the Nuo-PBs in Fig. 5 (the old Fig. 4).

- In the discussion (lines 390-394), the authors suggest the importance of phyB phosphorylation/modifications in the PB formation. Then, please describe how phyB can be phosphorylated/modified in plants (probably by phyB's intrinsic kinase activity or other kinases).

Response:

As the reviewer knows, the mechanism of phyB phosphorylation is still not fully understood. We feel that the mechanism of phyB phosphorylation per se is not closely relevant to the current stage of the investigation, but the fact that phosphorylation of phyB affects the stability of phyB and PB formation has been well documented.

- This study used a constitutively active mutant of phyB, named YHB. Recently, there is another report of constitutively active phyB mutant (named AtYVB in Jeong et al., Plant Physiol., 171: 2826-2840). Is there any difference between YHB and AtYVB (in terms of phyB function and nuclear localization)? If the two mutants are different, it would be interesting to compare the PB formation with YHB and YVB.

Response:

We particularly chose YHB, because it has been extensively studied both structurally and functionally. So, we could use this mutant to infer relationships between phyB conformation and PB dynamics. It would certainly be interesting to include more mutants in the future, including YVB.

- Check typo-errors in the manuscript. For example, line 121 ("four"), line 376 ("Tte"), and line 373 ("thermostabilitis"), and so on.

Response:

We have made the suggested changes.

- Check also reference format (especially titles, some have capital letters in each word).

Response:

We have edited the references.

Reviewer #3

The work by Hahm et al investigates the effect of temperature on the sub-nuclear localization of the light and temperature receptor phytochrome B (phyB). Using fluorescence microscopy, they describe the abundance and size distribution of phyB-containing nuclear bodies (photobodies). Although it had already been described that temperature has an effect on the size distribution of photobodies in plants grown in white light (Legris et al Science 2016), this paper shows that when plants grow in constant red light the temperature effects on photobodies is different from the previously reported temperature effects on photobodies in white light. Under continuous red light illumination, higher temperatures, which decrease the proportion of phyB in its active form, reduce the abundance of large photobodies while increasing their volume. This occurs both in cotyledons and hypocotyls and in two ecotypes, namely Landsberg erecta and Columbia. The authors notice that in many nuclei some photobodies are closer to the nucleolus and these frequently remain present in conditions where the average number of photobodies decreases. Shade also decreases the proportion of phyB in its active form but has a different effect on the size distribution of photobodies, which has been described previously (Trupkin et al., Plant Physiol 2014), and the authors confirm this finding and further analyze this phenomena in terms of association with the nucleolus. The main novelty of the work is the identification of a subset of photobodies that have higher

stability in high temperatures and are associated to the nucleolus. I have the following suggestions/comments.

1) the only distinctive feature of temperature-stable photobodies is that they appear near the nucleolus in deconvoluted epifluorescence images of DAPI-stained cells. It is not shown whether they are actually associated with the nucleolus in a more direct way. The chance of finding a photobody near the nucleolus in a picture could be affected, for instance, by the size of the nucleolus (if it was larger in high temperature the chance of finding a photobody near it would be higher). Thus, I suggest that the authors measure the size of the nucleolus in the different temperature conditions to evaluate this possibility. Also the work could be improved including a nucleolar marker in the images and doing 3D reconstitutions of the nuclei to improve the description of the association between photobodies and nucleoli. The biological significance of the association to the nucleolus should be discussed.

Response:

The main conclusion is individual PBs vary in thermostability. Both the numbers of both Nuo- and nonNuo-PBs decreased with temperature increases, but there are less Nuo-PBs than nonNuo-PBs in general, and there is always a Nuo-PB that can persist at 27°C. The epidermal cells have relatively large nuclei with a diameter of 10 µm (compared with other cell types, such as the mesophyll cell). In these nuclei, the size of the nucleolus is much smaller than the size of the nucleus, as seen in the images. We defined the Nuo-PBs based on a 3D model, in which the Nuo-PBs are localized to the nucleolar periphery. Although our current data cannot conclude whether the Nuo-PBs are directly associated with nucleolar components, this localization pattern is unlikely to be random. For example, if the nucleolar localization is random, one would expect to see that the percentage of Nuo-PBs (among all PBs) should stay the same in all conditions. This is clearly not the case, we observed large changes (by temperature, ecotype, tissue/organ type) in the numbers of nonNuo-PBs and less changes in the numbers of Nuo-PBs, thereby the percentage of Nuo-PBs varies among the experiments. In addition, in the epidermal cells of *PBG* during the 21°C-to-27°C transition, the number of PBs went from 2 to 1, however, the percentage of cells with a Nuo-PBs stayed the same (Fig. 4). Moreover, in *YHB*, the number of cells with one Nuo-PB stayed the same in all temperatures. These results argue against the idea that Nuo-PBs are the result of random PB localization. We still don't know the mechanism of PB formation/initiation and the distinct functions of the Nuo- and nonNuo-PBs. A major contribution of this study is to raise these important questions for future investigations.

2) The response of photobodies to temperature is not the same as to shade. Although both responses had been evaluated independently (Legris et al 2016, Trupkin et al 2014) no direct comparison had been done. The authors point out that in their conditions in response to high temperature they don't detect the appearance of small photobodies, typically found in response to shade. However, as far as I understood from the methods section, the procedure to evaluate each condition was different. The authors note that fixation weakened the signal of very small photobodies and used fixed samples to evaluate temperature effects but unfixed samples to evaluate shade effects. If this is the case, then the comparison is not good, and at least for this figure the high temperature experiment should be repeated without fixation to allow a

fair comparison with the effect of shade. Also, to make a better comparison milder FR treatments should be tested, to reach similar Pfr levels to those obtained in continuous RL 10uE at 27°C (so far the only condition tested is 10uE RL vs 10uE RL + 10 uE FR which is expected to give a largely different PfrB/Ptot than 10 uE at 27°C). Also, in fig 5b the number of nucleolar associated and not associated photobodies is compared between shade, 21°C and 27°C. It should be noted that in the case of shade there are small and large photobodies mixed and given that the dynamics of these two types of photobodies is different probably they represent phyB in different states and thus they should be treated/analyzed separately. Could the authors at least comment on this?

Response:

We thank this reviewer for the comments. Regarding the concerns of fixed and unfixed samples, please see the response to Reviewer 2 question 3. We agree that different FR conditions should result in different PB patterns, the 1:1 R/FR ratio used in our study is similar to what was used in the study by Trupkin et al.¹⁰. And, another reason to choose this condition is that we have observed this PB pattern in a similar condition⁶, which showed mostly small foci. It is rather arbitrary to define large PBs and small foci, because we don't really know how to distinguish them other than merely size. Nonetheless, if we use 0.2 μm^3 as the cutoff between PBs and small foci, as described previously⁹, the vast majority of the foci in shade were small foci, whereas the overwhelming majority in 27°C were PBs (Fig. 6c). These results therefore demonstrate that warm temperature and shade induce distinct PB dynamics. We added these in the last paragraph in the Results.

3) Another finding is that the abundance of photobodies and their response to temperature differs between cotyledons and hypocotyl and the number differs between the accessions Columbia and Landsberg erecta. For the first comparison, can the authors hypothesize whether this has any biological significance? Could it be merely due to differences in phyB protein levels among these cells? The comparison between accessions is a bit more difficult to interpret. One main concern I have is that the transgenes used are different among accessions. Not only the protein levels are different but also the fluorophores. I suggest to test whether the reported differences remain true using other transgenic lines of the same accession, ideally with different protein levels and fluorophores. Again, it would be good to test whether this difference is biologically significant. In this case I suggest to test it by evaluating hypocotyl elongation in response to temperature in the selected wild type and transgenic genotypes.

Response:

This is a good suggestion. We have characterized temperature responses in the hypocotyl and cotyledon in *Ler*, *Col-0*, *PBG*, and *PBC*. Please see our response to Reviewer 1 Question 2. It is unlikely that the difference in pattern between the two lines is caused by the difference between GFP and CFP, as these two fluorescent proteins differ only in a couple of amino acids. The level of phyB-GFP and phyB-CFP in the two transgenic lines are also comparable (Fig. 2f). We have added the following in the discussion:

“Our results show that the number of PBs varies in different tissue/organ types and *Arabidopsis* ecotypes (Fig. 2b and 3b,c). Interestingly, the variations in PB number by tissue/organ type, ecotype, and temperature are mainly contributed by nonNuo-PBs (Fig. 2b and 3b,c). Therefore, it

is tempting to speculate that the presence or absence of a PB, particularly those nonNuo-PBs, could account for tissue/organ- and ecotype-specific responsiveness to temperature. This point needs to be validated by future investigations on the role of individual PBs in tissue/organ-type and ecotype-specific light and temperature responses at the single cell level. It is unclear what determines the number of PBs. One hypothesis is that PB number is controlled by the stability of PBs, alternatively, it could also be determined by a seeding mechanism for initiating PB assembly and this seeding mechanism can be influenced by ecotype background and tissue/organ type.”

4) In the discussion the authors propose that phyB localized in photobodies associated to the nucleolus have a different kr2 than phyB in other photobodies. Based on the presented data this is a speculation and thus I think it shouldn't be in the figure. If the authors would like to test this hypothesis they could evaluate photobody dynamics using phyB mutated versions with normal photoconversion and different thermal stability (Zhang et al Plant physiol 2013, Burgie et al PNAS 2014) . An alternative to this model would be that the rate of phyB association to these nucleolar associated photobodies is increased by temperature (Rausenberger Plos One 2010 PMID: 20502669). To test this, it would be interesting to dissect the data of photobody volume into nucleolar associated and not associated photobodies (maybe it's not just that they remain but actually more phyB is being actively incorporated to the nucleolar associated photobodies in high temperatures). It should be discussed that it has been suggested that localization to photobodies inhibits dark reversion (Huang PNAS 2019 PMID: 30948632, Rausenberger Plos One 2010 PMID: 20502669, Klose New Phytol 2015 PMID: 260422).

Response:

We have toned down this particular hypothesis and focused our model (Fig. 6d) on the main conclusion that individual PBs have distinct thermostabilities. Please see our response to Reviewer 1 Question 1.

5) It should also be discussed that in the conditions tested, i.e. continuous red light and high temperature, hypocotyl length of Columbia and Landsberg erecta is regulated by a complex mechanism where increasing light intensities first inhibit elongation but then promote it (Johansson Nat commun 2014 PMID: 25258215). Although I understand that the aim of this paper is not to explain that process, it should be noted that in these conditions more phyB in the active conformation is not more efficiently promoting photomorphogenesis, and thus the biological significance of the abundance of photobodies is difficult to predict. Overall, the work could be improved by a description of a phyB-regulated developmental response (e.g. hypocotyl elongation) in the selected light and temperature conditions where the pattern of photobodies are described in order to get a better sense of the correlation between photobodies and a developmental response.

Response:

Great suggestions. We have performed phenotypic analysis on the temperature-dependent hypocotyl and cotyledon responses. Please see our response to Reviewer 1 Question #2. We do not think our experimental design allows us to test the temperature effect described by Johansson et al.. We added the following in the discussion: “It has also been shown that phyB promote

hypocotyl elongation at warm temperatures, however, our experiments were not particularly designed to test a correlation between PB dynamics and this particular response⁷⁰.”

6) Some final comments about the presentation of the results, in Fig 1c, 1d, 2d, 2e, 3b, 3c, 4d, the number of photobodies per cell is different (sometimes 1, sometimes 2). It would be easier to compare among conditions if all were representing the same values. Figure 2d and 2e show PBC and PBG in a different order from the rest of the figures and is confusing while comparing among figures.

Response:

Different criteria were used because the total number of PBs in the various lines or conditions were different. We picked the proper criteria to show the changes in the nuclei population -- i.e., if using static criteria, we might be able to capture the changes in a particular cell type or ecotype.

1. Qiu, Y., Li, M., Kim, R. J.-A., Moore, C. M. & Chen, M. Daytime temperature is sensed by phytochrome B in Arabidopsis through a transcriptional activator HEMERA. *Nat. Commun.* **10**, 140 (2019).
2. Balasubramanian, S., Sureshkumar, S., Lempe, J. & Weigel, D. Potent induction of Arabidopsis thaliana flowering by elevated growth temperature. *PLoS Genet.* **2**, e106 (2006).
3. Kumar, S. V. & Wigge, P. A. H2A.Z-containing nucleosomes mediate the thermosensory response in Arabidopsis. *Cell* **140**, 136–147 (2010).
4. Wigge, P. A. Ambient temperature signalling in plants. *Curr. Opin. Plant Biol.* **16**, 661–666 (2013).
5. Yoo, C. Y., Williams, D. & Chen, M. Quantitative Analysis of Photobodies. *Methods Mol. Biol.* **2026**, 135–141 (2019).
6. Chen, M., Schwab, R. & Chory, J. Characterization of the requirements for localization of phytochrome B to nuclear bodies. *Proc. Natl. Acad. Sci. U. S. A.* **100**, 14493–14498 (2003).
7. Chen, M., Tao, Y., Lim, J., Shaw, A. & Chory, J. Regulation of phytochrome B nuclear localization through light-dependent unmasking of nuclear-localization signals. *Curr. Biol.* **15**, 637–642 (2005).
8. Chen, M. *et al.* Arabidopsis HEMERA/pTAC12 Initiates Photomorphogenesis by Phytochromes. *Cell* vol. 141 1230–1240 (2010).
9. Van Buskirk, E. K., Reddy, A. K., Nagatani, A. & Chen, M. Photobody localization of phytochrome B is tightly correlated with prolonged and light-dependent inhibition of hypocotyl elongation in the dark. *Plant Physiol.* **165**, 595–607 (2014).
10. Trupkin, S. A., Legris, M., Buchovsky, A. S., Rivero, M. B. T. & Casal, J. J. Phytochrome B nuclear bodies respond to the low red to far-red ratio and to the reduced irradiance of canopy shade in Arabidopsis. *Plant Physiol.* **165**, 1698–1708 (2014).

REVIEWERS' COMMENTS:

Reviewer #1 (Remarks to the Author):

The revised manuscript by Hahm et al. is significantly improved by the addition of phenotypic analysis on the temperature-dependent hypocotyl and cotyledon responses. However, I feel that not all of the points raised in the previous round of review have been satisfactorily addressed by the authors.

I appreciate that the authors have now included data about temperature dependent physiological responses (inhibition of hypocotyl elongation and cotyledon expansion). Interestingly the data show that similar temperature-dependent PB dynamics in hypocotyl and cotyledon epidermal cells correlate with different temperature effects on phyB mediated growth phenotypes. The data are not easily interpretable and the authors could have invested more effort in unveiling the non-trivial correlation between PBs and physiological responses.

Line 160/161 "There was not an observable trend of the temperature effects on cotyledon expansion in 10 $\mu\text{mol m}^{-2} \text{s}^{-1}$ R light, despite some variations in cotyledon size between different temperatures (Fig. 1c, d)" I think it is also possible to argue that there is a slight trend for the wildtype at least from 16 – 27°C, where cotyledon size decreases, which would be in line with reduced phyB activity at higher temperature. The opposite effect (cotyledon area increased with temperature) was observed for lines overexpressing phyB-FP and wildtype grown in stronger (50 $\mu\text{mol m}^{-2} \text{s}^{-1}$) R light. Here it should be at least discussed that temperature could stimulate growth in light by mechanisms independent of the phyB inactivation by temperature (one of these mechanisms might still require phyB, as suggested by Legris Science 2016). The authors could have tested that for example using the BCY and YHB lines in light.

Line 159/160 "To our surprise, warmer temperatures further enhanced phyB-dependent cotyledon expansion in Ler and Col-0 in permissive 50 $\mu\text{mol m}^{-2} \text{s}^{-1}$ R light (Fig. 1c, d)" this is another interesting observation again pointing towards a possible phyB independent growth promoting mechanism that could also be potentially related to photosynthesis. The data are also in conflict with the PB pattern observed at 27°C under 50 $\mu\text{mol m}^{-2} \text{s}^{-1}$, because this indicated that the higher R intensity could compensate for the high temperature effect observed at 10 $\mu\text{mol m}^{-2} \text{s}^{-1}$ which is not the case for the hypocotyl growth inhibition.

Following up on this, the argumentation, that higher R intensity could compensate for the high temperature effect observed at 10 $\mu\text{mol m}^{-2} \text{s}^{-1}$ would be reasonable when PBs do not show temperature sensitivity anymore at 50 $\mu\text{mol m}^{-2} \text{s}^{-1}$. Unfortunately the authors only show PB patterns for 27°C, so this remains unclear.

The study thoroughly describes the PB patterns/dynamics in two different accessions and organs across the ambient temperature range. It further raises a number of important and interesting questions for further investigations. The authors repeatedly emphasize that the main conclusion is that individual PBs vary in thermostability and suggest that individual PBs could act as temperature sensors. However, the impact of the work would be substantially higher if some mechanistic insight into the observed phenomena could be provided.

With that said the study retains some weaknesses regarding the discussion of the underlying mechanism. The authors speculate that (thermal) stability of Pfr:Pfr in the PB is the determinant for PB thermostability. This hypothesis would be valid for PB stability in darkness but not in light, photoconversion is considerably faster compared with Pfr:Pfr thermal reversion. Further it is in conflict to the current model that warm temperature in the light affects Pfr:Pfr indirectly by accelerating the Pfr:Pr to Pr:Pr reversion (lines 464-466 and 500/501). To elucidate the mechanism the authors could have tested phyB variants with enhanced/reduced thermal stability. Additionally the authors could have addressed the alternative possibility that phyB has different binding affinities to individual PBs at the different temperatures, since it was shown that phyB constantly and quickly binds and unbinds to PBs, even in Pfr:Pfr.

Further, there is no particular discussion about how this relates to the idea that localization to photobodies inhibits dark reversion which was suggested by several studies including one from the Chen lab.

A minor comment: please check whether reference in lines 505 and 506 is incorrectly cited

Reviewer #2 (Remarks to the Author):

The authors (Hahm et al.) have resubmitted this manuscript and included a detailed response to the previous reviews, such as analysis of seedling phenotypes (hypocotyl length and cotyledon area) under different temperature conditions (i.e., newly added as Fig. 1 and Fig. 5e) and PB dynamics analysis under a high R-fluence rate (data included in Figs. 3 and 5), with improved writing of Introduction and description of results. In addition, I would accept the authors' responses for my previous comments. However, I have a couple of more comments to improve this manuscript.

1. There are many descriptions about the importance of N-terminal photosensory module for the thermostability of PBs: for example, lines 313-314 (the thermostability of PBs likely depends on phyB's N-terminal photosensory module), lines 374-375 (the thermostability of photobodies relies on phyB's N-terminal photosensory module), and lines 471-472 (our results showed that the thermosensitivity of PBs is determined by the N-terminal photosensory module of phyB). Then, I'm wondering why the authors did not investigate the PBs with transgenic lines expressing the N-terminal photosensory module (Since the transgenic lines have been reported by several research groups, I think the authors could obtain the lines easily or they might already have them). Thus, to conclude this more clearly, data of PBs observed with the transgenic lines expressing the N-terminal photosensory module need to be included in this study.

2. The results of the increases in phyB protein levels in warm temperature conditions were interesting to me (Figs. 2 and 4). For a possible explanation of these results, the authors suggested that the increase in phyB stability at warm temperatures might be due to the promoted conversion to the Pr form, which is more stable than the Pfr form. Another interesting result of this study might be the different PB dynamics between warm temperature and shade conditions (Fig. 6). Probably, these different PB dynamics might be related to the phenotypic differences (i.e., cotyledon expansion) between the two conditions. In the manuscript, it is also described that warm temperature in the light affects Pfr:Pfr indirectly by accelerating the Pfr:Pr to Pr:Pr reversion, whereas shade or FR directly photoconvert Pfr:Pfr to Pfr:Pr and Pr:Pr. If we consider a possibility of different protein stability of phyB dimers (Pfr:Pfr, Pfr:Pr/Pr:Pfr, and Pr:Pr), phyB protein levels in shade conditions might be different from those in warm temperature, which might explain the different PB dynamics. Thus, I'd like to ask the authors to compare the phyB protein levels under warm temperature and shade conditions.

Reviewer #3 (Remarks to the Author):

In this revised manuscript the authors included new experiments. However, the novelty and potential impact of the manuscript has not changed. The authors propose that different subsets of phytochrome B-containing nuclear bodies (PB) have different thermal sensitivities. The work remains descriptive and without any mechanistic insight on how these PB appear, disappear or remain unchanged.

I appreciate that the authors have modified their discussion and excluded their previously proposed model, which was lacking supporting data. Also, the new figure 1 addresses some of the comments I made in my first review, although the interpretation of the link between the effect of temperature on the morphological response and on the dynamics of PB is difficult to understand because the authors write "...elevated temperature inflicts opposite effects on phyB's functions in the hypocotyl and cotyledons..." (see for example abstract and lines 228-229), while their data shows that in continuous red light increasing the temperature leads to longer hypocotyls and larger cotyledons, see figure 1 where this is particularly obvious at R50 but even at R10 cotyledons

are larger at 27 than 12°C. In addition, as further explained below there is no direct evidence that the change in cotyledon size in continuous red light with increasing temperature in phyB mediated. I still consider that the work is of limited novelty, which contrast with the way the manuscript is written. This is to a significant extent due to incomplete referencing of previous work. Moreover, several claims made in the title and in the abstract are not well supported by the data. For example, the disassembly of phyB from photobodies (PBs) claimed in the title is one possibility explaining their data. However, others such as fusions of PBs at high temperature could also explain their data, a possibility that is not even mentioned. Finally, many concerns raised in the first revision were disregarded.

After the revision, the following major points remain problematic:

1. The novelty of this work relies on the identification of a subset of PB that have different temperature sensitivities. But these PB can only be identified by their position relative to the nucleolus. Whether this localization reflects some interaction with the nucleolus, or if it has any biological significance, or if these PB share any other characteristic such as their composition is not studied in this work. It is possible that the observed association with the nucleolus is stochastic and somehow alters the stability of those PBs but this change is not due to differences in the PBs themselves. I suggested alternatives to test this in the first review (comment 1) which were disregarded. Also, it should be stressed that the nucleolar-associated PB (Nuo-PB) are not temperature-insensitive, as it is stated in the text (e.g. lines 249-250; lines 398-399). Their number is also reduced by higher temperatures, although with a smaller slope than nonNuo-PB (Figures 2b, 3b). Finally, it has not been evaluated whether temperature actually disassembles PB. An alternative could be that PBs fuse at higher temperature which is also consistent with a lower number but a bigger PB volume (Fig. 2e, 3f).
2. In response to my comments and those of the other reviewers the authors measured hypocotyl elongation and cotyledon expansion in response to temperature. This is an improvement since it adds a new dimension to the work. However, I do not agree on the interpretation of the results. The authors consider that the cotyledon response in the wild type and transgenic lines is the same, but actually in the conditions where they do most microscopic analyses (10 μ mol.m⁻².s⁻¹) they differ clearly (Fig 1c, d). Also, the authors say that this temperature effect on cotyledon expansion is phyB-dependent but to my knowledge this is not clear in the literature and the authors do not provide any data to show that. I agree that the red light-induced cotyledon expansion was shown to be at least partially phyB mediated. However, to what extent phyB controls cotyledon expansion at a given red light intensity with changing temperatures would have to be shown. The difference between the effect of temperature on phyB activity or PB pattern and cotyledon expansion could be due to a phyB-independent mechanism of response to temperature. It was indeed shown previously that the temperature effect on growth can only be partially attributed to phytochrome (Legris et al., 2016). Also, in figure 5e the authors measure cotyledon expansion in response to temperature in dark-grown BCY seedlings. I understand that this makes sense for the YHB line but for the BCY line this experiment should be done in Rc to be compared with other lines.
3. The conditions used in this work are very particular and this should be stated (former comment 5). The advantage of this approach is that one can look more specifically at the effect of phyB. However, the trade-off is that the potential to draw broader conclusions to more realistic conditions is more limited. In this regard it is also intriguing that in figure 1a and 1b the authors show that seedlings grown at higher light intensities respond more to temperature than those grown at lower light intensities. This is unexpected and could be due to the specific conditions chosen (continuous red light).

Other comments

1. The fact that PB respond to temperature with a different pattern than they respond to shade was shown previously (Legris et al 2016).
2. I still have concerns about comparing fixed and not fixed samples (former comment 2).
3. The fact that according to modeling studies localization to PB inhibits thermal reversion should be at least mentioned (former comment 4).

Response to Reviewers

We wanted to thank all the reviewers for their valuable suggestions and comments.

Reviewer #1

The revised manuscript by Hahm et al. is significantly improved by the addition of phenotypic analysis on the temperature-dependent hypocotyl and cotyledon responses. However, I feel that not all of the points raised in the previous round of review have been satisfactorily addressed by the authors.

I appreciate that the authors have now included data about temperature dependent physiological responses (inhibition of hypocotyl elongation and cotyledon expansion). Interestingly the data show that similar temperature-dependent PB dynamics in hypocotyl and cotyledon epidermal cells correlate with different temperature effects on phyB mediated growth phenotypes. The data are not easily interpretable and the authors could have invested more effort in unveiling the non-trivial correlation between PBs and physiological responses.

1. Line 160/161 “There was not an observable trend of the temperature effects on cotyledon expansion in 10 $\mu\text{mol m}^{-2} \text{s}^{-1}$ R light, despite some variations in cotyledon size between different temperatures (Fig. 1c, d)” I think it is also possible to argue that there is a slight trend for the wildtype at least from 16 – 27°C, where cotyledon size decreases, which would be in line with reduced phyB activity at higher temperature. The opposite effect (cotyledon area increased with temperature) was observed for lines overexpressing phyB-FP and wildtype grown in stronger (50 $\mu\text{mol m}^{-2} \text{s}^{-1}$) R light. Here it should be at least discussed that temperature could stimulate growth in light by mechanisms independent of the phyB inactivation by temperature (one of these mechanisms might still require phyB, as suggested by Legris Science 2016). The authors could have tested that for example using the BCY and YHB lines in light.

Response:

We thank this reviewer for the comment. We revised the sentence to “No linear trend was observed for the temperature effects on cotyledon expansion in 10 $\mu\text{mol m}^{-2} \text{s}^{-1}$ R light, despite a slight reduction of cotyledon size from 16°C to 27°C.” We also revised the text in Discussion (the 4th paragraph) to “despite the opposite temperature effects, both hypocotyl and cotyledon cells showed similar PB dynamics, suggesting that the discrepancy in the temperature-dependent hypocotyl and cotyledon responses might be due to tissue/organ-specific signaling circuitry either downstream of PB dynamics or independent of phyB¹.” We agree that the temperature-dependent cotyledon expansion is quite interesting, and it also requires careful genetic analysis using not only the transgenic lines described here (for photobody analysis) but also phytochrome signaling mutants to dissect the genetic mechanism. Given the scope of the current study on temperature-dependent photobody dynamics, we think those experiments should be included in a separate study.

2. Line 159/160 “To our surprise, warmer temperatures further enhanced phyB-dependent cotyledon

expansion in *Ler* and *Col-0* in permissive 50 $\mu\text{mol m}^{-2} \text{s}^{-1}$ R light (Fig. 1c, d)” this is another interesting observation again pointing towards a possible *phyB* independent growth promoting mechanism that could also be potentially related to photosynthesis. The data are also in conflict with the PB pattern observed at 27°C under 50 $\mu\text{mol m}^{-2} \text{s}^{-1}$, because this indicated that the higher R intensity could compensate for the high temperature effect observed at 10 $\mu\text{mol m}^{-2} \text{s}^{-1}$ which is not the case for the hypocotyl growth inhibition. Following up on this, the argumentation, that higher R intensity could compensate for the high temperature effect observed at 10 $\mu\text{mol m}^{-2} \text{s}^{-1}$ would be reasonable when PBs do not show temperature sensitivity anymore at 50 $\mu\text{mol m}^{-2} \text{s}^{-1}$. Unfortunately the authors only show PB patterns for 27°C, so this remains unclear.

Response:

The light-dependent cotyledon expansion has been shown by Neff and Van Volkenburgh to be *phyB*-dependent almost 30 years ago². They showed cotyledon expansion in monochromatic red light requires higher light intensity and this response was lost in the *phyB* mutant (*hy3*) as well as in the *phyB* chromophore-deficient mutant *hyl*, providing convincing genetic evidence supporting the notion that cotyledon expansion is mediated by *phyB* signaling. The contrasting effects of *phyB* in promoting cotyledon expansion by inhibiting hypocotyl elongation have long been recognized. More recently, it was shown that one of the reasons for this discrepancy is due to the differential responses of a group of PIF-regulated SAUR genes in the hypocotyl and cotyledon³. We agree that the current data could not exclude the possibility of a *phyB*-independent effect on cotyledon expansion, and therefore, pointed out this possibility in the conclusion in the revised manuscript.

3. The study thoroughly describes the PB patterns/dynamics in two different accessions and organs across the ambient temperature range. It further raises a number of important and interesting questions for further investigations. The authors repeatedly emphasize that the main conclusion is that individual PBs vary in thermostability and suggest that individual PBs could act as temperature sensors. However, the impact of the work would be substantially higher if some mechanistic insight into the observed phenomena could be provided. With that said the study retains some weaknesses regarding the discussion of the underlying mechanism. The authors speculate that (thermal) stability of Pfr:Pfr in the PB is the determinant for PB thermostability. This hypothesis would be valid for PB stability in darkness but not in light, photoconversion is considerably faster compared with Pfr:Pfr thermal reversion. Further it is in conflict to the current model that warm temperature in the light affects Pfr:Pfr indirectly by accelerating the Pfr:Pr to Pr:Pr reversion (lines 464-466 and 500/501). To elucidate the mechanism the authors could have tested *phyB* variants with enhanced/reduced thermal stability. Additionally the authors could have addressed the alternative possibility that *phyB* has different binding affinities to individual PBs at the different temperatures, since it was shown that *phyB* constantly and quickly binds and unbinds to PBs, even in Pfr:Pfr. Further, there is no particular discussion about how this relates to the idea that localization to photobodies inhibits dark reversion which was suggested by several studies including one from the Chen lab.

Response:

In the 2nd submission, we particularly tried to avoid claiming that it is the stability of the Pfr:Pfr

determines the thermostability of photobodies. We revised our model in Fig. 6D by only showing the thermostability of individual photobodies are different, which is observed by our experiments. The fact that the nuclear bodies of phyB's C-terminal module (BCY) did not respond to temperature variations suggests that the thermostability of photobodies is determined by the N-terminal photosensory module. This conclusion is further supported by the YHB mutant because YHB lacks the Pfr and Pr interconversion in the N-terminal module. We added a sentence in the Discussion regarding the photobody effect on the thermal reversion of phyB: "Photobody-localization of phyB has been suggested to stabilize the Pfr form or attenuate thermal reversion⁴, our results therefore suggest that individual photobodies might stabilize Pfr to different extents and thereby causing the distinct temperature requirements for phyB disassembly among PBs – i.e., individual PBs carry distinct thermosensitivities (Fig. 6d)."

4. A minor comment: please check whether reference in lines 505 and 506 is incorrectly cited

Response:

We changed the reference to Legris et al. 2016.

Reviewer #2

The authors (Hahm et al.) have resubmitted this manuscript and included a detailed response to the previous reviews, such as analysis of seedling phenotypes (hypocotyl length and cotyledon area) under different temperature conditions (i.e., newly added as Fig. 1 and Fig. 5e) and PB dynamics analysis under a high R-fluence rate (data included in Figs. 3 and 5), with improved writing of Introduction and description of results. In addition, I would accept the authors' responses for my previous comments. However, I have a couple of more comments to improve this manuscript.

1. There are many descriptions about the importance of N-terminal photosensory module for the thermostability of PBs: for example, lines 313-314 (the thermostability of PBs likely depends on phyB's N-terminal photosensory module), lines 374-375 (the thermostability of photobodies relies on phyB's N-terminal photosensory module), and lines 471-472 (our results showed that the thermosensitivity of PBs is determined by the N-terminal photosensory module of phyB). Then, I'm wondering why the authors did not investigate the PBs with transgenic lines expressing the N-terminal photosensory module (Since the transgenic lines have been reported by several research groups, I think the authors could obtain the lines easily or they might already have them). Thus, to conclude this more clearly, data of PBs observed with the transgenic lines expressing the N-terminal photosensory module need to be included in this study.

Response:

It has been shown previously by both Akira Naganati⁵ and us^{4,6} that phyB's N-terminal module itself does not localize to photobodies, the C-terminal module is required and sufficient for photobody localization. We showed recently that photobody localization of phyB relies on the dimerization of the histidine-kinase-related domain within the C-terminal module⁷.

2. The results of the increases in phyB protein levels in warm temperature conditions were

interesting to me (Figs. 2 and 4). For a possible explanation of these results, the authors suggested that the increase in phyB stability at warm temperatures might be due to the promoted conversion to the Pr form, which is more stable than the Pfr form. Another interesting result of this study might be the different PB dynamics between warm temperature and shade conditions (Fig. 6). Probably, these different PB dynamics might be related to the phenotypic differences (i.e., cotyledon expansion) between the two conditions. In the manuscript, it is also described that warm temperature in the light affects Pfr:Pfr indirectly by accelerating the Pfr:Pr to Pr:Pr reversion, whereas shade or FR directly photoconvert Pfr:Pfr to Pfr:Pr and Pr:Pr. If we consider a possibility of different protein stability of phyB dimers (Pfr:Pfr, Pfr:Pr/Pr:Pfr, and Pr:Pr), phyB protein levels in shade conditions might be different from those in warm temperature, which might explain the different PB dynamics. Thus, I'd like to ask the authors to compare the phyB protein levels under warm temperature and shade conditions.

Response:

We thank this reviewer for the suggestion. Shade also increases the level of phyB, as it converts Pfr to Pr, which is usually being stabilized. This was shown in this study by Wang *et al.*⁸. So, the photobody dynamics in shade is not due to a decrease in the level of phyB. I do agree that the observation of warm-temperature-dependent increases in the phyB level is quite intriguing, it would certainly be interesting to investigate the cause of this response in the future.

Reviewer #3

In this revised manuscript the authors included new experiments. However, the novelty and potential impact of the manuscript has not changed. The authors propose that different subsets of phytochrome B-containing nuclear bodies (PB) have different thermal sensitivities. The work remains descriptive and without any mechanistic insight on how these PB appear, disappear or remain unchanged.

Response:

We disagree with this assessment of our study. How signaling molecules (i.e., the photoreceptor in this case) in a signal transduction pathway respond to the signal at the single cell level is a critical part of the cellular signaling mechanism. We would argue that the understanding of the cell biology aspects of signaling remains to be the weakest links of many signaling pathways in both animal and plant models. Mechanistic insight of the biochemical or genetic interactions among signaling components are important, however, the information usually missing is where the described signaling events occur in the cell, whether the events occur in the same cell, and whether they occur the same way in all cell types. These questions regarding the cellular mechanisms of signaling require quantitative investigation of the behaviors of the signaling molecules during cell signaling events. Such cell biology data complement the molecular genetic and biochemical studies, and therefore, could either support or argue against the currently accepted signaling mechanisms. We were excited and surprised by the new finding of this study because the temperature-dependent photobody dynamics argue against the prevailing model of temperature signaling -- the gradual decreases in photobody number with increases in

temperature were particularly surprising to us (and obviously to the reviewers as well) because these could not be predicted or are even counter intuitive based on the current models. Although we still do not understand how individual photobodies respond to temperature differently, the novel findings from this study raised this new important question and opened the way to investigate the distinct functions of individual photobodies in cell-type specific responses, thereby setting the foundation to understanding the mechanism of phytochrome signaling at the single-cell level.

I appreciate that the authors have modified their discussion and excluded their previously proposed model, which was lacking supporting data. Also, the new figure 1 addresses some of the comments I made in my first review, although the interpretation of the link between the effect of temperature on the morphological response and on the dynamics of PB is difficult to understand because the authors write "...elevated temperature inflicts opposite effects on phyB's functions in the hypocotyl and cotyledons..." (see for example abstract and lines 228-229), while their data shows that in continuous red light increasing the temperature leads to longer hypocotyls and larger cotyledons, see figure 1 where this is particularly obvious at R50 but even at R10 cotyledons are larger at 27 than 12°C. In addition, as further explained below there is no direct evidence that the change in cotyledon size in continuous red light with increasing temperature in phyB mediated. I still consider that the work is of limited novelty, which contrast with the way the manuscript is written. This is to a significant extent due to incomplete referencing of previous work. Moreover, several claims made in the title and in the abstract are not well supported by the data. For example, the disassembly of phyB from photobodies (PBs) claimed in the title is one possibility explaining their data. However, others such as fusions of PBs at high temperature could also explain their data, a possibility that is not even mentioned. Finally, many concerns raised in the first revision were disregarded.

After the revision, the following major points remain problematic:

1. The novelty of this work relies on the identification of a subset of PB that have different temperature sensitivities. But these PB can only be identified by their position relative to the nucleolus. Whether this localization reflects some interaction with the nucleolus, or if it has any biological significance, or if these PB share any other characteristic such as their composition is not studied in this work. It is possible that the observed association with the nucleolus is stochastic and somehow alters the stability of those PBs but this change is not due to differences in the PBs themselves. I suggested alternatives to test this in the first review (comment 1) which were disregarded. Also, it should be stressed that the nucleolar-associated PB (Nuo-PB) are not temperature-insensitive, as it is stated in the text (e.g. lines 249-250; lines 398-399). Their number is also reduced by higher temperatures, although with a smaller slope than nonNuo-PB (Figures 2b, 3b). Finally, it has not been evaluated whether temperature actually disassembles PB. An alternative could be that PBs fuse at higher temperature which is also consistent with a lower number but a bigger PB volume (Fig. 2e, 3f).

Response:

We agree with this reviewer that this work has raised many new interesting questions, such as why individual PBs respond to temperature differently. Although we tried to distinguish the individual PBs using the nucleolus as a marker, obviously, new subnuclear landmarks need to be

developed to label individual PBs. In our previous response to the reviewer comments, we laid out reasons why the nucleolus-association is unlikely to be stochastic. If it were stochastic, one should have observed the same percentage of Nuo-PBs in all nuclei, this is clearly not the case. We did not conclude that Nuo-PBs are temperature-insensitive, a subset of them are clearly temperature-sensitive as we showed in Fig. 2 and 3. However, in the cotyledon nuclei of *PBG*, because there were only two PBs, the particular Nuo-PB was temperature-insensitive.

We have mainly entertained the idea that warm-temperature-dependent destabilization of the Pfr triggers PB disassembly because PBs are the foci where the active Pfr phyB localizes (based on genetic evidence and computational modeling). If only the PB number is considered, it is possible, in theory, that the reduction of PB number by temperature is caused by PB fusion, but, based on our understanding of the nature of PB formation, this is unlikely because PB fusion would require further stabilization of the Pfr, and therefore, the photobody-fusing model is inconsistent with the findings that warm temperatures destabilize the Pfr^{1,9}.

2. In response to my comments and those of the other reviewers the authors measured hypocotyl elongation and cotyledon expansion in response to temperature. This is an improvement since it adds a new dimension to the work. However, I do not agree on the interpretation of the results. The authors consider that the cotyledon response in the wild type and transgenic lines is the same, but actually in the conditions where they do most microscopic analyses (10 μmol.m-2.s.-1) they differ clearly (Fig 1c, d). Also, the authors say that this temperature effect on cotyledon expansion is phyB-dependent but to my knowledge this is not clear in the literature and the authors do not provide any data to show that. I agree that the red light-induced cotyledon expansion was shown to be at least partially phyB mediated. However, to what extent phyB controls cotyledon expansion at a given red light intensity with changing temperatures would have to be shown. The difference between the effect of temperature on phyB activity or PB pattern and cotyledon expansion could be due to a phyB-independent mechanism of response to temperature. It was indeed shown previously that the temperature effect on growth can only be partially attributed to phytochrome (Legris et al., 2016). Also, in figure 5e the authors measure cotyledon expansion in response to temperature in dark-grown BCY seedlings. I understand that this makes sense for the YHB line but for the BCY line this experiment should be done in Rc to be compared with other lines.

Response:

The light-dependent cotyledon expansion has been shown by Neff and Van Volkenburgh to be phyB-dependent almost 30 years ago. They showed that cotyledon expansion in monochromatic red light requires higher light intensity (i.e., above 50 μmol m⁻² s⁻¹ R light) and that this response was lost in the *phyB* mutant (*hy3*) as well as in the phyB chromophore-deficient mutant *hy1*, providing convincing genetic evidence supporting the notion that cotyledon expansion is mediated by phyB signaling². We agree that the current data could not absolutely exclude the possibility of a phyB-independent effect, so we have revised our conclusion by pointing out that this could also be due to a phyB-independent mechanism. Please also see our response to the first question of Reviewer #1.

3. *The conditions used in this work are very particular and this should be stated (former comment 5). The advantage of this approach is that one can look more specifically at the effect of phyB. However, the trade-off is that the potential to draw broader conclusions to more realistic conditions is more limited. In this regard it is also intriguing that in figure 1a and 1b the authors show that seedlings grown at higher light intensities respond more to temperature than those grown at lower light intensities. This is unexpected and could be due to the specific conditions chosen (continuous red light).*

Response:

We agree with this comment and have added the following sentence in the Discussion: “It is important to point out that the PB responses shown in monochromatic R light or R light supplemented with FR light could be modulated in the natural light conditions wherein multiple classes photoreceptors are in action.”

Other comments

1. *The fact that PB respond to temperature with a different pattern than they respond to shade was shown previously (Legris et al 2016).*

Response:

We discussed in the Discussion that what we observed in this study is quite different from what was reported in Legris *et al.* 2016. These are likely due to the different experimental settings: steady-state patterns in this study vs. patterns during temperature transitions (Legris *et al.* 2016)¹, specific cell types in this study vs. two cell layers in Legris *et al.*, and 3D-quantification of photobody patterns vs. a 2D method in Legris *et al.*

2. *I still have concerns about comparing fixed and not fixed samples (former comment 2).*

Response:

The reason why we switched to the fixed method is because we know that, based on our experience, using fresh samples is more easily to introduce errors due to the quick responses of phyB-FP to the excitation light during confocal or fluorescence microscopy. Although we just published the method paper on this protocol, this is not the first time we are using this method, we have used fixed samples for photobody analysis in the following publications: Huang et al (2016)¹⁰, Qiu et al (2017)⁷, Yoo et al (2019)¹¹, and Yang et al (2019)¹².

3. *The fact that according to modeling studies localization to PB inhibits thermal reversion should be at least mentioned (former comment 4).*

Response:

Please see our response to the 3rd question of Reviewer #1.

1. Legris, M. *et al.* Phytochrome B integrates light and temperature signals in Arabidopsis. *Science* **354**, 897–900 (2016).
2. Neff, M. M. & Van Volkenburgh, E. Light-Stimulated Cotyledon Expansion in Arabidopsis Seedlings (The Role of Phytochrome B). *Plant Physiol.* **104**, 1027–1032 (1994).
3. Sun, N. *et al.* Arabidopsis SAURs are critical for differential light regulation of the

- development of various organs. *Proc. Natl. Acad. Sci. U. S. A.* **113**, 6071–6076 (2016).
4. Van Buskirk, E. K., Reddy, A. K., Nagatani, A. & Chen, M. Photobody localization of phytochrome B is tightly correlated with prolonged and light-dependent inhibition of hypocotyl elongation in the dark. *Plant Physiol.* **165**, 595–607 (2014).
 5. Matsushita, T., Mochizuki, N. & Nagatani, A. Dimers of the N-terminal domain of phytochrome B are functional in the nucleus. *Nature* **424**, 571–574 (2003).
 6. Chen, M., Tao, Y., Lim, J., Shaw, A. & Chory, J. Regulation of phytochrome B nuclear localization through light-dependent unmasking of nuclear-localization signals. *Curr. Biol.* **15**, 637–642 (2005).
 7. Qiu, Y. *et al.* Mechanism of early light signaling by the carboxy-terminal output module of Arabidopsis phytochrome B. *Nat. Commun.* **8**, 1905 (2017).
 8. Quail, P. H., Martinez-Garcia, J. F. & Devlin, P. F. A novel high-throughput in vivo molecular screen for shade avoidance mutants identifies a novel phyA mutation. *Journal of* (2011).
 9. Jung, J.-H. *et al.* Phytochromes function as thermosensors in Arabidopsis. *Science* **354**, 886–889 (2016).
 10. Huang, H. *et al.* PCH1 integrates circadian and light-signaling pathways to control photoperiod-responsive growth in Arabidopsis. *Elife* **5**, e13292 (2016).
 11. Yoo, C. *et al.* Phytochrome activates the plastid-encoded RNA polymerase for chloroplast biogenesis via nucleus-to-plastid signaling. *Nat. Commun.* **10**, 2629 (2019).
 12. Yang, E. J. *et al.* NCP activates chloroplast transcription by controlling phytochrome-dependent dual nuclear and plastidial switches. *Nat. Commun.* **10**, 2630 (2019).